# Using regulatory variants to detect gene–gene interactions identifies networks of genes linked to cell immortalisation

D. Wragg[1], Q. Liu[1], Z. Lin[1], V. Riggio[1], C.A. Pugh[1], A.J. Beveridge[2], H. Brown[1], D.A. Hume[3], S.E. Harris [4], I.J. Deary[4], A. Tenesa [1] & J.G.D. Prendergast[1]*

The extent to which the impact of regulatory genetic variants may depend on other factors, such as the expression levels of upstream transcription factors, remains poorly understood. Here we report a framework in which regulatory variants are first aggregated into sets, and using these as estimates of the total cis-genetic effects on a gene we model their non-additive interactions with the expression of other genes in the genome. Using 1220 lymphoblastoid cell lines across platforms and independent datasets we identify 74 genes where the impact of their regulatory variant-set is linked to the expression levels of networks of distal genes. We show that these networks are predominantly associated with tumourigenesis pathways, through which immortalised cells are able to rapidly proliferate. We consequently present an approach to define gene interaction networks underlying important cellular pathways such as cell immortalisation.

---

[1] The Roslin Institute, University of Edinburgh, Easter Bush, Midlothian EH25 9RG, UK. [2] Glasgow Polyomics, College of Medical, Veterinary and Life Science, University of Glasgow, Glasgow, UK. [3] Mater Research Institute-University of Queensland, Translational Research Institute, Woolloongabba, QLD 4102, Australia. [4] Centre for Cognitive Ageing and Cognitive Epidemiology, University of Edinburgh, Edinburgh EH8 9JZ, UK. *email: James.Prendergast@roslin.ed.ac.uk

    

Differences in gene expression levels between individuals underlie substantial variation in complex traits and diseases among human populations[1], and arise due to a combination of differences in the genome of individuals (genetic effects) and the environment they are exposed to (environmental effects)[2]. Understanding how these factors combine, and potentially interact, to shape a gene's expression level can improve our understanding of the biological mechanisms underlying important phenotypes and disease risk. However, there remain large gaps in our knowledge of how each gene's expression level is determined, partly due to the complexity of gene regulation[3]. Each gene may be under the direct or indirect control of dozens of genetic cis and trans regulatory loci, and transcription and environmental influences, but how these interact to shape each gene's expression levels is still poorly understood.

Consortia such as GTEx[4] and GEUVADIS[5] have generated substantial amounts of data on expression quantitative trait loci (eQTLs) in the human genome. However, these studies have predominately focused on the effect of individual variants near to genes, cis-eQTLs, in isolation from their wider cellular environment[6,7]. This is partly because exhaustive scans that include the study of distal effects are burdened by stringent multiple testing corrections. Each cis-eQTL has, though, been found to generally only explain a small proportion of the variation in a gene's expression between individuals[8]. This suggests that for most genes there remains a large amount of variation unaccounted for.

It seems likely that the effect of cis-eQTLs may often depend on the expression of upstream genes, higher up in the same pathway, or, as many eQTLs are thought to act by disrupting the binding site of transcription factors (TFs)[9], the effect of the eQTL may depend on the expression level of the corresponding TF. Non-additive interactions between the eQTL and these other genes' expression levels may explain some of the uncaptured variation of the gene's expression between individuals. Zhernakova et al.[10] identified many eQTLs whose size of effect was related to the expression of distal genes, with the overwhelming majority related to cell type composition differences between the samples. Each human genome, though, contains millions of genetic variants and more than twenty thousand genes, and consequently a comprehensive scan for trans gene–eQTL interactions involves a punitive multiple testing burden which is a particular issue in scans for non-additive interactions where effect sizes are generally smaller. Attempts to reduce this search space have focused on interactions with known TF[11] or co-expressed genes[12], under the assumption that the expression of an upstream regulator will to some extent correlate with the expression of the gene in question. Previous studies[7,13] have also illustrated how various forms of interactions can lead to genotype-dependent variance in expression and have restricted their analyses to genes displaying such effects. These studies have though predominantly focused on non-additive interactions between nearby eQTLs.

Another option to reduce the search space without having to restrict the analysis to specific gene sets or cis effects is to first aggregate the effects of all the cis-genetic variants for each gene into regulatory variant-sets. A given gene may be associated with a number of cis-regulatory variants and its expression determined by the combination of alleles carried across these sites. By modelling the expression of a gene according to an individual's cis-regulatory variants, as demonstrated by Gamazon et al.[8], it is possible to capture the total variation in a gene's expression explained by known cis-genetic effects, improving the amount of expression variability explained between individuals. Identifying interactions with this cis-genetic component involves far fewer tests than testing each underlying genetic variant in turn, and would not require any assumptions to be made regarding which genes may be involved. A previous study illustrated how the cis-genetic component modelled in this way can be correlated to the expression level of distal genes to identify directed gene regulatory networks[14]. However, non-additive interactions between the combined cis-genetic effects of genes and the expression of distal transcripts is largely unexplored.

Epstein-Barr Virus (EBV) is responsible for around 200,000 cases of cancer annually[15] and the immortalisation of B cells to generate lymphoblastoid cell lines (LCLs) has proven to be a good model for investigating the mechanisms underlying EBV-associated cancers[16]. EBV expresses a number of latency genes both in cancers and LCLs that drive cellular immortalisation[16]. These include EBV nuclear antigens (EBNAs) which are TFs that target both viral and host genes, as well as the latent membrane protein LMP1 which activates host NF-κB TFs, such as p50, p52, RelA, RelB and cREL ultimately driving lymphoid cell proliferation, differentiation and survival[17].

In this study we model the expression of each human gene in LCLs according to their local genetic variation and attempt to identify non-additive interactions between this cis-genetic component of the gene's expression levels and the expression of other genes in the genome. Using an independent replication dataset on a different expression assay we show that this approach can potentially identify novel gene–gene interactions missed by other approaches, providing mechanistic insights into human gene regulation and the networks of genes linked to EBV-induced cellular immortalisation.

## Results

**Independent replication of gene expression prediction models.** Using lymphoblastoid cell line (LCL) gene expression and whole-genome sequencing (WGS) data across 876 Scottish individuals of European ancestry from the Lothian Birth Cohort 1936 (LBC1936, http://www.lothianbirthcohort.ed.ac.uk)[18,19], we trained models of gene expression levels from SNPs located between 1 Mb upstream of each gene's transcription start site and 1 Mb downstream of its termination site using PrediXcan[8]. These models provide a prediction of a gene's expression based solely on the cis-genetic variants an individual carries (Fig. 1a). The $R^2$ of the prediction models are largely uncorrelated to gene size ($R^2$ between gene size and prediction accuracy = 0.00167; Supplementary Fig. 1). For the 9,316 gene probes with significant prediction models, 4283 were removed due to excessive kurtosis in their observed expression values. From the remaining 5033 gene probes, 1387 (28%) with a prediction $R^2 \geq 0.1$, representing 1205 genes (Fig. 2a) were retained.

To see which of these prediction models replicated in an independent population, the same models were used to predict expression levels in a set of 344 European individuals from the 1000 Genomes (GEUVADIS) Consortium with matching LCL expression data. Despite the differences between these datasets, including the LBC1936 expression being measured with micorarrays rather than RNA-seq data, these new model predictions have an $R^2 \geq 0.1$ with the measured expression at 361 of the 1205 genes retained from the analysis of the LBC1936 dataset (Fig. 2b), of which 311 pass the kurtosis filter. These 311 genes with strong, reproducible, cis-regulatory genetic effects were retained in downstream analyses.

**Regulatory variant-set-dependent variance in gene expression.** The aim of this study was to identify whether the impact of the expression level of a distal gene, gene[B], on a gene, gene[A], potentially depends on gene[A]'s set of regulatory variants (as represented by its PrediXcan predicted expression levels; Fig. 1b). To do this we first investigated the evidence for uncaptured

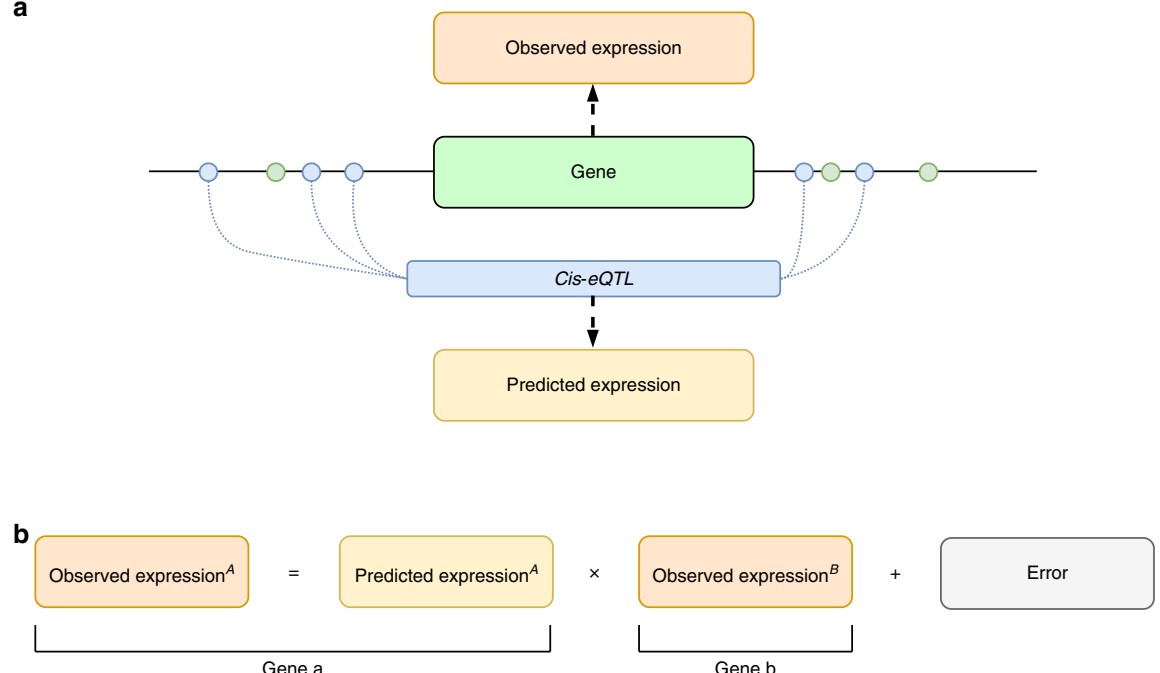

**Fig. 1 Employing a gene's cis-eQTL complement to identify genetic interactions. a** SNPs within 1 Mb of a gene's transcription start and termination sites were used to train an expression level prediction model using PredictDB. Predictions from these models correspond to the additive effect of all cis-regulatory variants for a given gene. **b** The observed expression of gene[A] can then be modelled as an interaction (X) between its cis-eQTL (predicted expression) and the observed expression of other genes in the genome (gene[B]). The error term represents, for example, uncaptured environmental or trans effects linked to variation in the expression of gene[A], not capture by these other terms.

interactions potentially underlying the expression of this set of 311 gene[A]s by mapping variance of expression quantitative trait loci (veQTL) linked to these genes. Previous studies have highlighted how genetic and environmental interactions can lead to genotype-dependent variance in a gene's expression[7,20]. We illustrate why this is the case using simulated data in Fig. 3. If the effect of an eQTL depends on a second factor, for example the expression level of a TF to which it binds, chromatin regulator or an upstream gene in the same pathway, then the gene's expression will only be high under the correct combination of eQTL genotype and second gene's expression level. As shown in Fig. 3b this interaction between the eQTL and second gene will lead to differences in the variance in the expression of the gene dependent upon the eQTL's genotype. Even if the interacting factor is unknown, the presence of genotype-dependent variance in expression levels can point towards a potential interaction contributing to the gene's expression level. Although traditionally the study of veQTL has been restricted to individual variants, as we show in Fig. 3d–f the same principle extends to sets of regulatory variants. If a gene is under the control of multiple cis-regulatory variants the same effect is still expected to be observed, with greater variance in expression levels depending on the corresponding set of regulatory variants an individual carries.

As each different predicted expression level in this analysis corresponds to a unique set of cis-genotypes we explored, using the LBC1936 dataset whether variance of expression differed by predicted expression levels across individuals, indicating the potential existence of uncaptured interactions with the cis-regulatory variant-sets. Using a non-parametric approach analogous to that of Brown et al.[7], but extended beyond single SNPs to sets of regulatory variants (see methods), we identify 129 genes (41%; Spearman's rho FDR <0.05) in the LBC1936 dataset showing evidence of variant-set-dependent variance in expression

levels when accounting for any eQTL effects. In the GEUVADIS dataset 201 of the 311 genes (65%) show set-dependent variance effects, of which 87 overlap those found in the LBC1936 analysis (Supplementary Data 1). Consequently, more than a third of all genes under strong cis-regulatory control show reproducible evidence of set-dependent variance in expression. For comparison, Brown et al.[7] found veQTL for 508 out of 13,660 genes, of which 23 are also present within the 87 genes identified in both of our datasets (Supplementary Data 1). An example veQTL found across both the LBC1936 and GEUVADIS datasets is shown in Fig. 4. The cis-regulatory variants of *SLFN5* (Schlafen family member 5) explain ~60–70% of its variation in expression levels between individuals. The variance in expression levels is, though, greater among those individuals carrying regulatory variant-sets linked to higher expression of this gene, as indicated by the association between its predicted expression levels and the square of the residuals after regressing out the eQTL effect (see methods). These results suggest that uncaptured interactions with these regulatory variant-sets potentially exist among a large proportion of these genes.

**Regulatory variant-set interactions with distal genes.** To identify if other genes potentially interacting with the cis-genetic components of these genes might explain the observed variant-set-dependent variance in expression, the predicted expression of each of the 311 genes (366 gene probes) was tested for a statistical interaction with the observed expression of every other gene in the genome, while accounting for sex, age and population structure (see Methods). A total of 8,268,610 probe pairs were tested in the LBC1936 dataset, of which 237,291 display evidence for a significant interaction (ANOVA F-test FDR <0.05). Of these gene–gene interactions 3953 (across 7,052 probe pairs) replicate

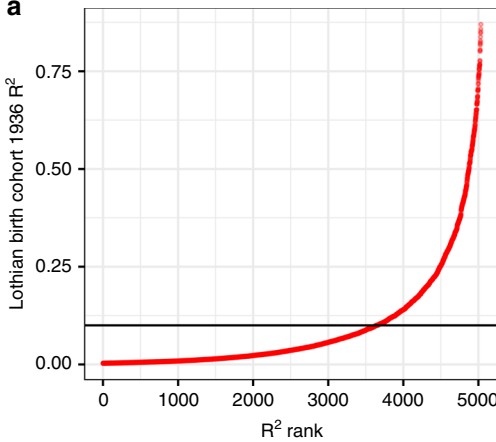

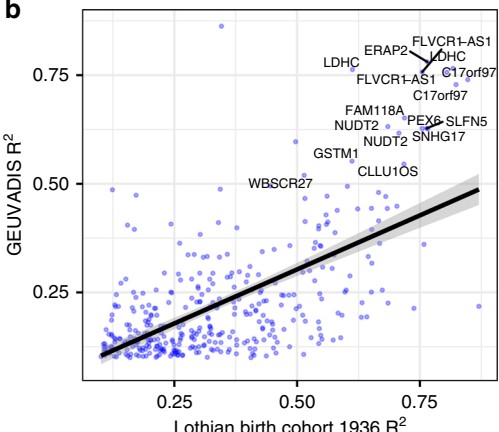

**Fig. 2 Model prediction performance and validation. a** $R^2$ of predicted versus observed gene expression levels in the Lothian Birth Cohort (LBC1936) against the rank of the $R^2$ value. Only the 5033 genes that passed the kurtosis threshold are shown. The horizontal line indicates an $R^2 = 0.1$, for which the 1205 genes exceeding this threshold were retained for further analysis. **b** Validation of the LBC1936 gene prediction models using the 1000 Genomes data. Genes with an $R^2 > 0.1$ across both datasets are plotted, with genes with an $R^2 > 0.5$ in both datasets labelled. Genes annotated twice have more than one probe on the array used in the LBC1936 dataset.

As shown in Fig. 5, the patterns of expression we observe are consistent with those expected from the simulated data (Fig. 3). In this example, the previously observed set-dependent variance in expression of *SLFN5* (Fig. 4a) is linked to the expression of a second distal gene, *DANCR*, with the expression of *SLFN5* only highest on certain cis-regulatory backgrounds when the observed expression of *DANCR* is also low. *DANCR* is a long non-coding RNA, directly regulated by MYC[21], that is associated with promoting cancer cell proliferation[22], whereas *SLFN5* is a tumour suppressor whose expression is negatively correlated to cancer cell invasiveness[23].

A possible limitation of combining the effects of multiple cis-genetic variants into a single prediction score is the potential for masking interactions associated with a single regulatory variant. To investigate this we repeated the above analyses but restricting the prediction models to just the lead eQTL for each gene. 262 unique genes (306 probes) have prediction models with an $R^2 > 0.1$ in both the LBC1936 and GEUVADIS datasets. Of these, 239 genes are also present in the 311 that pass the corresponding filters in the variant-set analysis. As shown in Supplementary Fig. 2 prediction accuracies are generally substantially higher in the variant-set analysis, with the single best eQTL models largely only having higher $R^2$ where both models have comparatively poor prediction accuracies. Repeating the interaction tests with these single best eQTL prediction models highlights that although 45.5% of interactions are detected in both analyses, 36.1% are only identified when using the variant-set models. Notably a subset of interactions (18.4%) are only identified when using the single best eQTL models, suggesting that the variant-set approach leads to the larger number of identified interactions, but some interactions are missed, potentially due to masking effects.

A large proportion of the 2472 interactions identified using the variant-set models are centred on the same genes (Supplementary Fig. 3). The genes in these networks show little evidence of clustering in the genome, with the majority (93%) of interactions involving genes on different chromosomes. Despite the observed significant statistical interactions the majority of gene pairs also show little evidence of being co-expressed, with a mean Pearson's $r$ between the expression of genes in each pair across individuals of only $0.023 \pm 0.038$ standard deviations (S.D.) in the LBC1936 and $0.028 \pm 0.04$ in the GEUVADIS dataset. Importantly, of the different genes whose observed expression interacts with the predicted expression/regulatory variant-set of the same gene, few show evidence of being co-expressed (Supplementary Fig. 4). Gene co-expression has previously been used to identify pairs of genes potentially falling within the same expression network[12]. We consequently tested whether the interacting gene pairs were generally co-expressed. Correlation tests performed on the observed expression of the 4699 probe pairs, return 3614 with significant co-expression in either dataset, while 1908 are significant across both datasets (Pearson's correlation FDR < 0.05; LBC1936 = 3,127, GEUVADIS = 2395; Supplementary Data 7). However, correlation estimates are low within these significantly co-expressed pairs, with a mean $r^2$ of just $0.033 \pm 0.043$ in LBC1936 and $0.051 \pm 0.046$ in GEUVADIS. This suggests that restricting the analysis to co-expressed genes as in previous studies would have potentially missed a large number of the putative interactions identified in the current study.

Genotype-dependent variance in expression is thought to be a potential marker of non-additive interactions. Of the 87 genes (96 probes) initially identified as showing evidence of set-dependent variance in expression in both datasets, 48 genes (54 probes) are associated with an interaction between their set of regulatory variants and the observed expression of at least one other gene. This is significantly more than expected by chance (Fisher's exact test $P = 1.21 \times 10^{-16}$), supporting the hypothesis

in the GEUVADIS dataset following multiple testing correction (Supplementary Data 2), with 2821 (5185 probe pairs, i.e., 74%) showing a consistent direction of effect between datasets, significantly more than expected by chance (comparison of sign concordance of the interaction coefficient between datasets among significant gene–gene interactions versus all gene pairs tested $P < 2.2 \times 10^{-16}$, $\chi^2 = 1948.2$, $df = 3$). To minimise the impact of potential statistical artefacts, gene pairs where the interaction might also be explained by a nearby genetic variant (12 unique pairs in LBC1936 and 7 pairs in GEUVADIS, 19 unique pairs in total; Supplementary Data 3 and 4) or the expression of a third gene (271 pairs in LBC1936, 109 pairs in GEUVADIS, 364 unique pairs in total—of which 7 were previously identified as having interacting SNPs, 2 in LBC1936, 5 in GEUVADIS; Supplementary Data 5 and 6) were removed, leaving 2472 reproducible, putative regulatory variant-set by gene expression interactions (across 4699 probe pairs) where the impact of a gene's set of regulatory variants are associated with the expression of a distal gene (summarised in Supplementary Table 1).

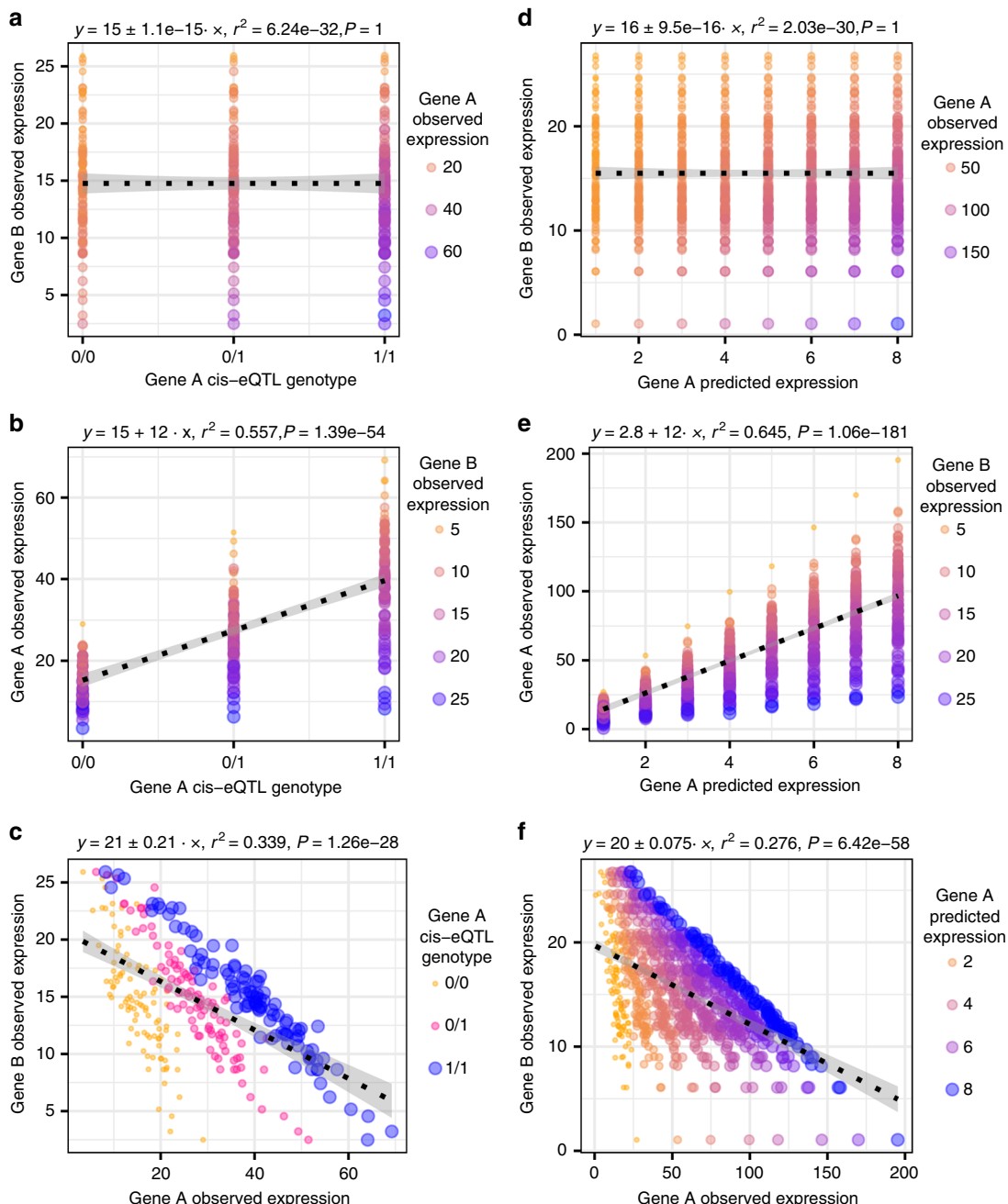

**Fig. 3 Interactions can lead to genotype-dependent and regulatory variant-set-dependent variance in a gene's expression levels.** Simulated data illustrating that **a** gene A's expression is only highest when the cis-eQTL genotype is 1/1 and the expression level of a distal gene (gene[B]) is low. **b** Replotting the data in **a** on different axes illustrates how this interaction is associated with genotype-dependent variance in the expression of gene[A], with greater variance in expression between individuals when the eQTL genotype is 1/1. **c** The interaction leads to genotype-dependent differences in the slope of the relationship between gene[A] and gene[B]'s expression levels. **d**–**f** The same principle extended to sets of cis-regulatory variants. P-values derived from Spearman's rho test.

that set-dependent variance in expression can be used to prioritise genes potentially linked to these interactions. However, 33 genes that do not display evidence of set-dependent variance also show evidence of interactions between their predicted expression and the expression of a second gene, for example *OXTR:BHLHE22* (Supplementary Fig. 5A). Although *OXTR* displays evidence of genotype-dependent variance in expression in the GEUVADIS dataset it is not significant in the LBC1936 dataset (Supplementary Fig. 5B). This suggests that although the identification of veQTLs is an effective approach to prioritise

putative interactions, a third of those identified in this study would have been missed if pre-filtering by the presence of veQTL as in other studies.

**Little evidence genetic interactions drive the associations.** As a gene's observed expression level is a combination of cis and trans genetic effects as well as environmental factors we investigated if any of these interactions could be explained by interactions just between the two genes' cis-regulatory variants i.e., excluding the

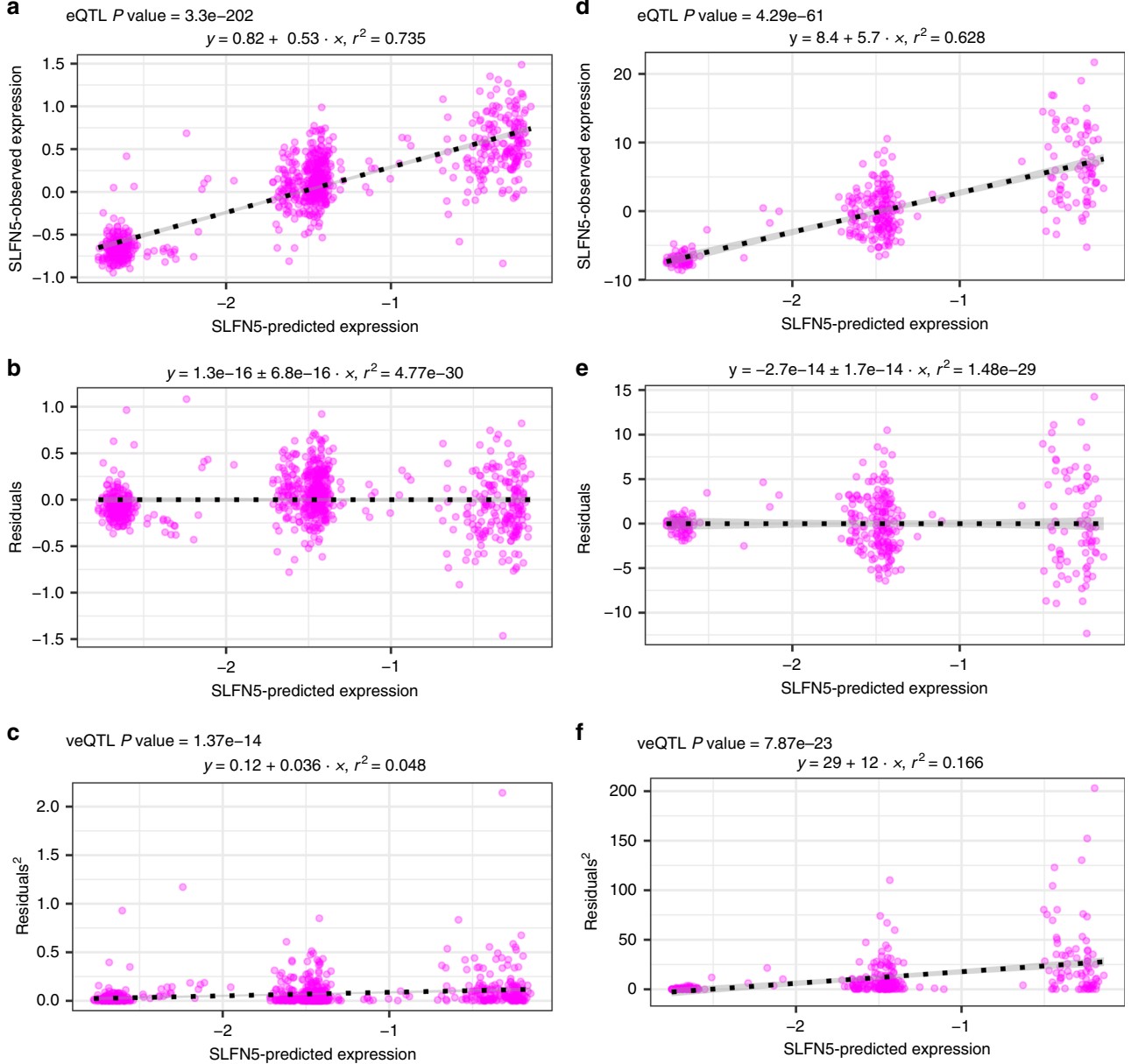

**Fig. 4 Detecting regulatory variant-set veQTL.** An example regulatory variant-set-based veQTL identified at the *SLFN5* locus is shown. **a** *SLFN5* shows a strong correlation between its predicted and observed expression levels in the Lothian Birth Cohort suggesting this gene is under the strong control of nearby regulatory variants. **b** Plotting the residual after regressing out this cis-eQTL effect against predicted expression levels suggests that this gene also shows set-dependent variance in expression. **c** Correlating the gene's predicted expression level to the square of the residuals from **b** confirms the presence of a veQTL at this locus. **d**–**f** the same as **a**–**c** but resulting from analysis of the 1000 Genomes data. *P*-values derived from Spearman's rho test.

potential trans genetic and environmental effects associated with gene[B]. To do this we fitted interactions between the predicted expression of both genes. From this analysis 28 pairs are significant in the LBC1936 dataset (ANOVA F-test FDR < 0.05; Supplementary Data 8) of which only one, *FAM167A:PLCG2* (FDR = 0.028), replicates in the GEUVADIS dataset (FDR = 0.026; Supplementary Fig. 6).

To explore this putative genetic interaction further we investigated which interactions remained after accounting for the cis-genetic component of the second gene's expression, i.e., whether any interactions could be completely explained by the identified eQTLs. To do this, we repeated the test for an interaction between the predicted expression of the first gene and the observed expression of the second, but also accounted for the

genotypes at all of the cis-regulatory variants identified by PrediXcan for the second gene. By controlling for the effects of the nearby regulatory variants we were able to test whether the cis-genetic variants for gene[B] could completely explain the observed interaction. The statistical interaction is no longer observed in 80 gene pairs (35 in LBC1936, 50 in GEUVADIS), of which 5 gene pairs are common to both datasets (Supplementary Data 9). The previously identified interaction based on cis-regulatory variants, *FAM167A:PLCG2*, remains significant in this analysis (ANOVA F-test; LBC1936 FDR = $4.59 \times 10^{-05}$, df = 397; GEUVADIS FDR = 0.0056, df = 264). These analyses suggest that the links between these gene pairs are not predominantly driven by genetic interactions between their cis-regulatory variants.

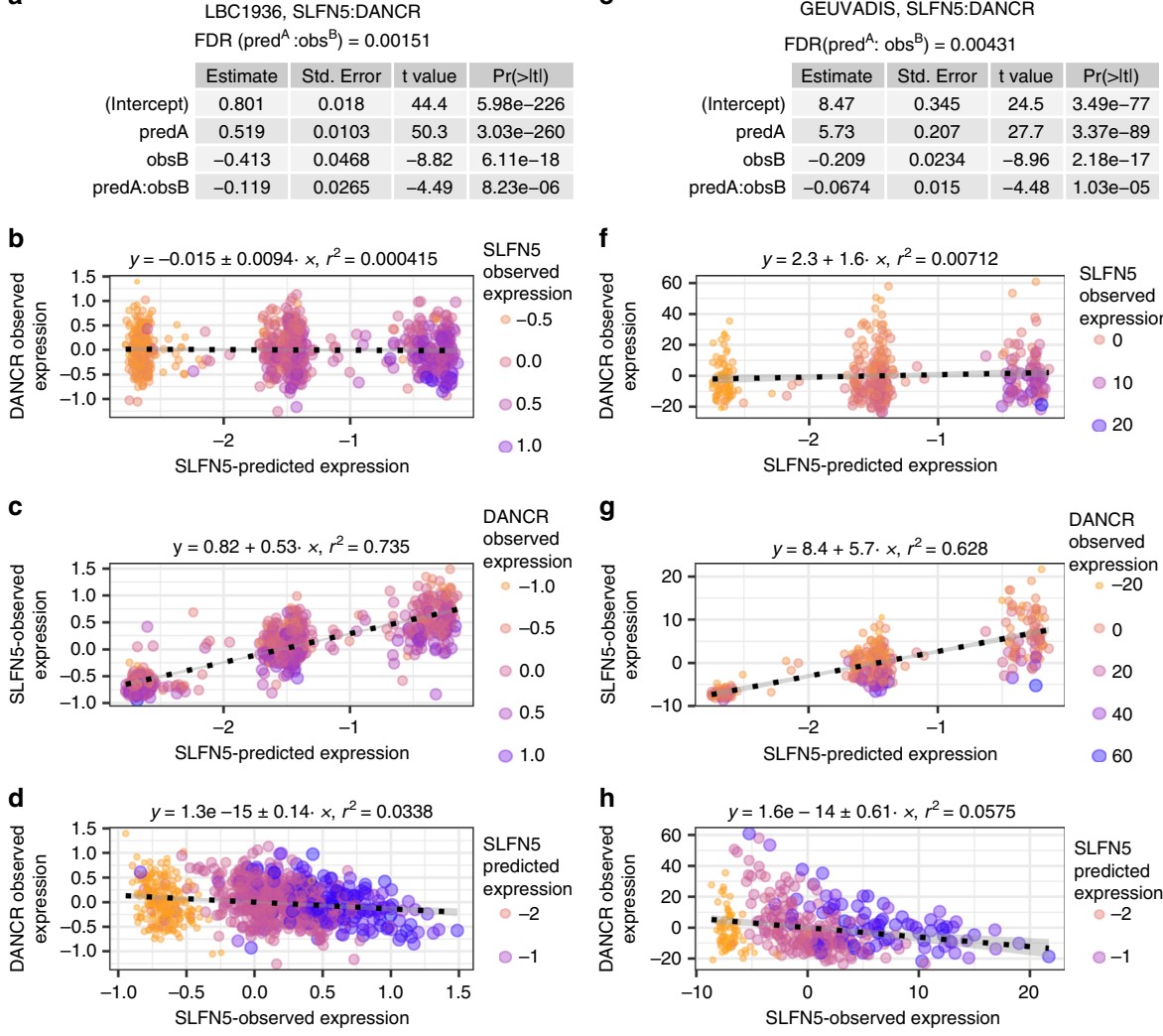

**Fig. 5 Significant regulatory variant-set-dependent interaction between *SLFN5* and *DANCR*.** Example of a significant set-dependent interaction, showing patterns consistent with the simulated data presented in Fig. 3. The expression of *SLFN5* is linked to the combination of cis-regulatory variant-set and expression of *DANCR*. This is shown to be the case for the Lothian Birth Cohort (**a**–**d**) and is reproduced in the 1000 Genomes data (**e**–**h**), with both datasets exhibiting statistical significance (FDR < 0.05) and sign concordance for the interaction term coefficient. *P* and FDR values derived from ANOVA F-test.

**The interaction networks are linked to cell immortalisation.** The predicted expression of *LDHC* (lactate dehydrogenase C), involved in anaerobic glycolysis and ATP synthesis, shows a significant statistical interaction with the expression of 360 genes. Interactions with multiple factors is potentially indicative of a hidden strong interaction with an untested trans factor driving the pathway as a whole[11]. The genes interacting with *LDHC* are enriched for the mitochondrion organisation (hypergeometric test FDR = $8.3 \times 10^{-10}$) and oxidative phosphorylation (hypergeometric test FDR = $9.9 \times 10^{-3}$) GO terms (Supplementary Table 2, Supplementary Data 10), indicating they are linked to the same ATP metabolic pathways as *LDHC* and likely reflect underlying biological links. Likewise, the neurogenesis GO term is enriched among the genes showing a statistical interaction with the *SYNGR1* gene that is associated with presynaptic vesicles in neuronal cells. *SLFN5*, an interferon inducible immune gene[24], is preferentially associated with genes linked to interferon response, and *FLVCR1-AS1*, that has been shown to sponge microRNAs[25] is associated with the targets of miR-21. This association with genes involved in shared pathways suggests these observed links are not spurious associations.

A common theme to these networks is a strong link to cancer and cell immortalisation. In primary cells the expression of *LDHC* is largely restricted to the testes[26], but the gene has been shown to be reactivated in proliferating tumour cells[27,28] to enable ATP synthesis via aerobic glycolysis. *SLFN5* has been shown to promote tumourigenesis in glioblastomas[29] and *FLVCR1-AS1* knockdown in hepatocellular carcinomas inhibits cell proliferation[25]. Analysing the interacting genes as a whole, as well as strong associations with genes linked to the *BRCA1* tumour suppressor and *MYCN* oncogene, one of the strongest enrichments is for genes downregulated in nasopharyngeal carcinomas, a form of cancer strongly associated with the EBV transformation of epithelial cells[30]. The association of these networks with EBV transformation is supported by other enriched terms such as targets of the histone methyltransferase *EZH2* which is linked to epigenetic regulation in EBV-transformed B cells[31].

To explore the links between these networks and EBV immortalisation we examined the binding to the promoters of these interacting genes of various key EBV latency proteins and NF-κB subunits. As illustrated in Fig. 6 the interacting genes in these networks are strongly enriched with binding of the EBV

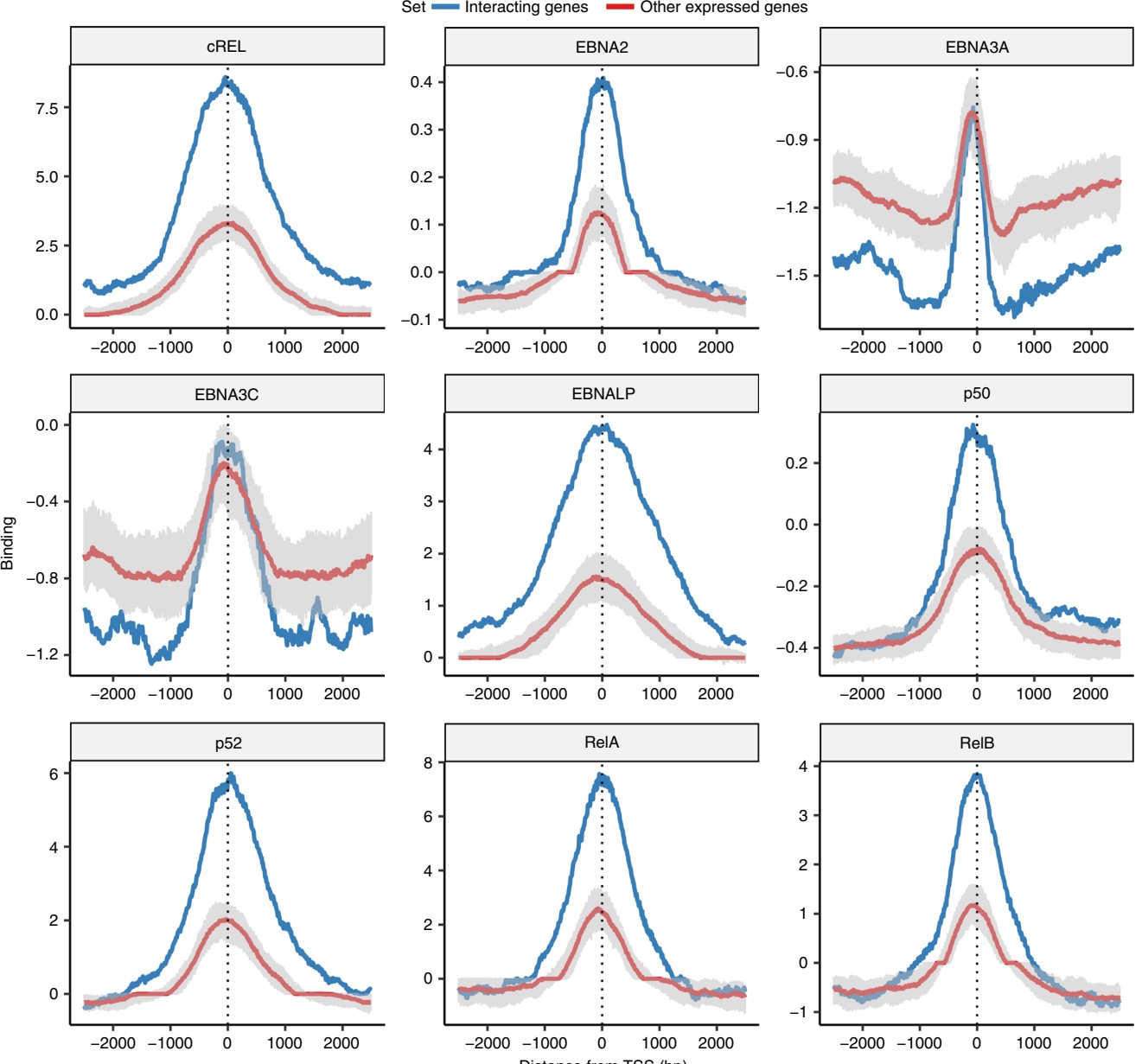

**Fig. 6 Binding of key EBV-associated transcription factors at the promoters of genes in the interaction networks.** The median binding levels of four EBV nuclear antigens (EBNA2, EBNA3A, EBNA3C and EBNALP) and five NF-κB subunits (cREL, p50, p52, RelA and RelB) at the promoters of interacting genes (i.e., interacting gene[B]s. Blue line) as well as genes expressed and tested for an interaction but for which one was not observed (red line). The grey shaded area around the red line represents the 95% confidence interval calculated by sampling 100 times the same number of genes from this background set as were in the foreground set of interacting genes. The processed binding data used in this plot was obtained from Jiang et al.[16].

transcriptional coactivators EBNA2 and EBNALP at their promoters when compared to other genes expressed in these cell lines but not members of these networks. NF-κB subunits, including relA and p52, that are induced by the EBV latent membrane protein LMP1[16], are also found to be bound at 3–4 times the levels at these interacting genes. This suggests that these networks at least in part reflect gene pathways targeted by the EBV transformation machinery.

**Additional variance explained by the observed interactions.** We finally quantified how much additional expression variance is explained by the interactions identified. On average (median) a given interaction explains an additional 0.68% ± 0.98 S.D. of the variation in a gene's expression in the LBC1936 dataset and 2.0% ± 1.3 in the GEUVADIS dataset above and beyond the additional variance explained by the main effects of the interaction gene (Fig. 7; Supplementary Data 11). However, as the genes in each network show little evidence of co-expression, this suggest that each is potentially explaining different variation in the expression of the central gene. To explore this, stepwise regression was used to identify the set of independent gene[B]'s for each gene[A] (having 2 or more interactions; Supplementary Data 12). On average the set of non-redundant distal genes explains a further 13.2% ± 9.7 (S.D.) of the variation in a gene[A]'s expression in the LBC1936 and 17.6% ± 11.0 in the GEUVADIS datasets,

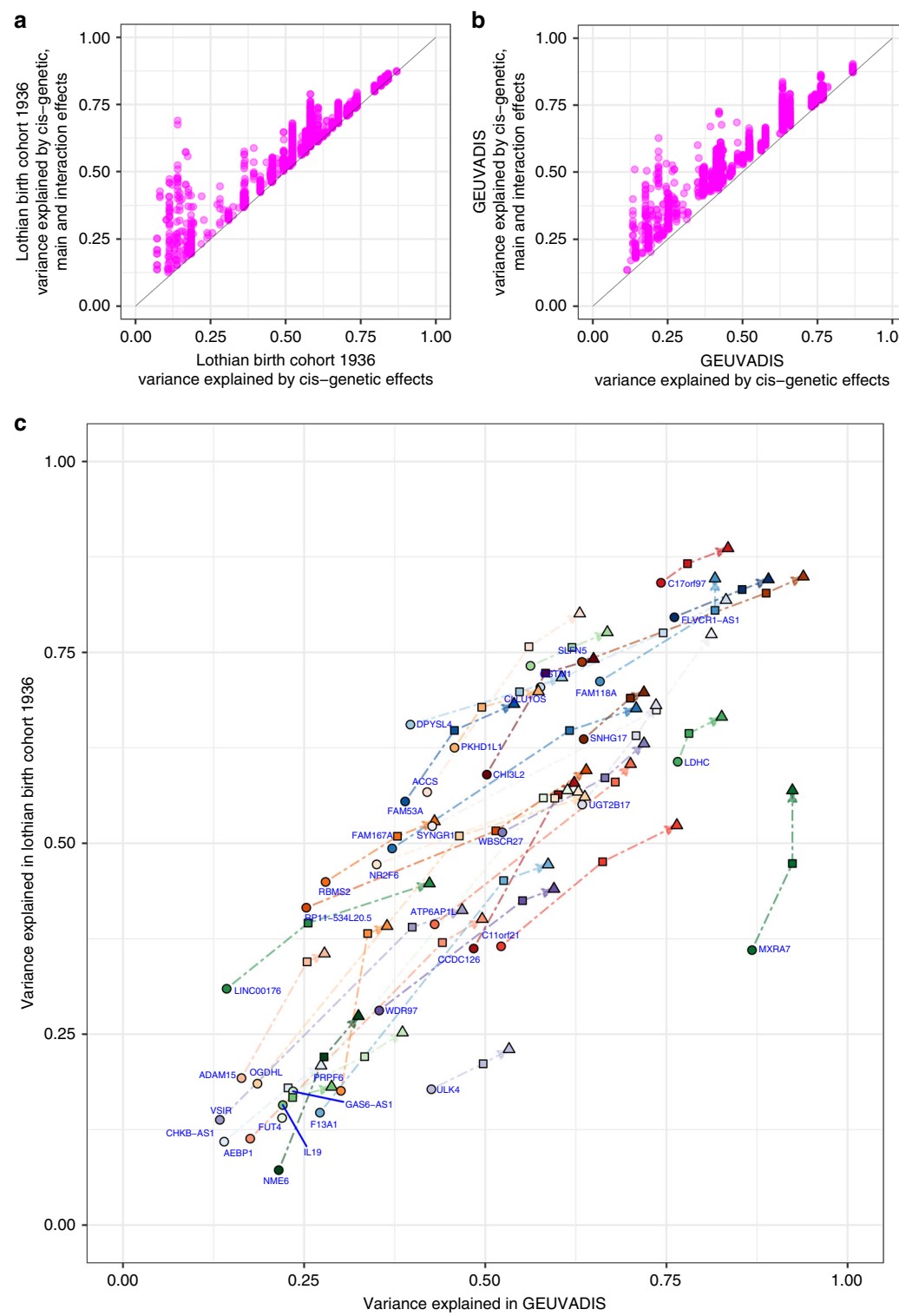

**Fig. 7 The proportion of variance explained by additive and interaction effects between cis-regulatory variants and the expression of distal genes.** A given interaction explains on average **a** an additional 0.68% of the variation in the gene's expression in the Lothian Birth Cohort, and **b** an additional 2.0% in the GEUVADIS dataset. **c** The total set of non-redundant interacting genes identified by stepwise regression explained 13.2% of the variation in gene expression in the Lothian Birth Cohort and 17.6% in the GEUVADIS dataset, with the interaction effects alone explaining 2 and 5% of the variation on average respectively (points are coloured by gene name).

with the interaction effects alone (i.e., excluding the main effects) explaining 2% ± 2.2 and 5% ± 3.6 of the variation, respectively. The correlation between the variance explained for each gene is high between datasets (Fig. 7c) with the exception of *MXRA7* where the variance explained by both the cis-genetic and main trans expression effects observed in the LBC1936 are less than half that observed in the GEUVADIS dataset. It is possible this results from greater technical variation in the measurement of this gene's expression in the LBC1936 microarray compared to the GEUVADIS sequencing data. Consequently, although each individual interaction explains a relatively small proportion of expression variance, collectively they can explain a substantial proportion of the variation in a gene's expression levels not captured by eQTLs alone.

## Discussion

In this study we aimed to identify gene–gene interactions using WGS and gene expression data, and to cross-validate these using an independent dataset of a different age profile to reduce false positives. Castaldi et al.[11] highlighted two important methodological issues when screening for interaction effects between eQTLs and the expression of distal genes: (1) relates to the pre-processing of data to account for variability across samples, libraries, or experimental conditions, which typically includes a normalisation step to obtain near-Gaussian data; (2) relates to statistical issues arising in the presence of moderate to strong interaction effects that can induce substantial heterogeneity of variance by genotype class, and spurious interactions. The approach we have taken in this study attempts to minimise the first of these issues by cross-validating the interactions identified by the gene expression microarray data using an independent population and method of gene expression quantification, mRNA sequencing. As RNA microarray and sequence data are normalised using different methods, the reproducibility of the interactions across these datasets should reduce the likelihood of scale-dependent effects. Another advantage of replication across these two datasets is that although the LBC1936 cohort is a set of unusually old individuals well matched for age and geographic location, the GEUVADIS cohort is comparatively younger so that any interactions specific to the unusually elderly nature of the first cohort are less likely to replicate.

Although the second issue, i.e., that a strong interaction between the regulatory variants and an unmeasured factor may lead to spurious associations with the factors that are tested, is difficult to completely exclude, we ensured that nearby variants and other genes could not be driving the observed associations. The genes being returned falling within the same pathways suggests that the interactions are not spurious associations, but, as in any study based on correlations between factors, it is not possible to exclude the possibility that a factor sitting upstream of all the genes in the observed networks is the true interacting factor with the observed interactions showing a strong link to the processes driving EBV immortalisation.

The majority of interactions identified in this study involved genes on different chromosomes. The largest networks each contained in excess of 100 genes, with the largest centred around *LDHC*. This is a germ-cell specific gene (www.proteinatlas.org)[32], typically expressed only in the testis, however, multiple splice variants of the gene have also been found to be significantly expressed in a wide range of tumours including melanoma, breast, colon, prostate, lung, renal, ovarian, thyroid and cervical cancers[27]. Lactate dehydrogenase catalyzes the final step in anaerobic glycolysis through the conversion of pyruvate to lactate[33]. A high glycolytic rate is favourable for proliferating cells, facilitating the use of glucose to produce high levels of ATP[34].

LCLs have been found to produce high levels of lactate, lactate dehydrogenase and pyruvate, suggesting that activation of the aerobic glycolytic pathway also occurs in EBV-transformed lymphoblastoid cells, corresponding to a phenomenon known as the Warburg effect[35]—in which proliferating cells tend to favour aerobic glycolysis[36]. There is increasing interest in the possibility of targeting the Warburg effect as a potential therapeutic target in cancers[37]. However, there are many outstanding questions as to how and why tumour cells use the Warburg effect[38] and consequently LCLs may provide a convenient and readily accessible model for this process and the potential role of gene–gene interactions.

The second largest cluster was centred on *MXRA7* and was enriched for interacting genes observed to be upregulated in prostate cancer cells following knockdown of the transcriptional repressor *NIPP1*. *NIPP1* is involved in the maintenance of the H3K27me3 methylation mark by *EZH2* in proliferating cells, and more generally the total list of all interacting genes was observed to be substantially enriched for genes down-regulated following the knockdown of *EZH2*. *EZH2* is upregulated following EBV infection[31], leading to the epigenetic repression of tumour suppressor genes, and *GSK126*, has been proposed as a potential drug targeting EBV latency due to its induction of cell cycle arrest and apoptosis[39] via the potent and specific inhibition of *EZH2*. Consequently some of the signals detected in this study may reflect the widespread chomatin changes attributable to the epigenetic regulator, *EZH2*, that is targeted by the EBV nuclear antigens following infection[39].

The network centred on *SLFN5* was enriched with genes linked to interferon gamma upregulation and viral infection. As the IFN-gamma cytokine is linked to resistance to viral infection it is down-regulated by EBV proteins[40]. Although the individual context-dependent eQTLs previously identified by Zhernakova et al.[10] were predominantly linked to cell type composition differences they also observed evidence for a set of cis-eQTLs dependent on interferon signalling pathways.

Consequently, the approaches in this study highlight networks of genes linked to EBV-induced cell transformation. We illustrated how restricting to genes displaying variant-set-dependent variation in expression levels would be an effective approach at reducing the testing burden of this kind of analysis. This approach has the advantage that it is not necessary to have prior biological evidence of genes interacting, for example due to them falling into the same pathway or being co-expressed, allowing for further interactions to be detected compared to other prioritisation approaches. There are however limitations to this approach, primarily that if the gene is not under reasonably strong control of cis-genetic regulatory variants it is not tested. Also, we show that although the variant-set approach identifies the most interacting gene pairs, a subset are only identified when using the single best eQTL, likely due to masking effects linked to individual eQTLs. A hybrid analysis combining both these approach may be most effective at identifying the maximum number of interaction pairs. As transcript levels do not necessarily equate to protein levels fitting the protein levels of transcription factors may better capture extra variation in the expression of genes.

We cross-referenced our genes with significant veQTL against those identified by several independent studies[7,41,42]. Specifically, we identified 12 genes in common (*BBS2*, *C17orf97*, *CLLU1OS*, *DPYSL4*, *FUT4*, *LDHC*, *MXRA7*, *NMNAT3*, *PAX8*, *SERPINB10*, *TIMM10*, *WBSCR27*) with the TwinsUK LCL results of Wang et al.[42,43]. Cross-referencing Brown et al.'s[7] peak veQTL in the same TwinsUK cohort, we identified 23 genes in common (*ACCS*, *CHI3L2*, *CMAHP*, *CRIPAK*, *DPYSL4*, *ERAP1*, *FAM53A*, *FAM118A*, *FAM167A*, *GAA*, *GJC1*, *PARD6G*, *PAX8*, *PIP5K1C*, *PLA2G4C*, *POLR1E*, *SAMD10*, *SERPINB10*, *TLE6*, *WBSCR27*, *ZNF239*,

*ZNF266, ZNF880*). Finally, we cross-referenced the cis- specific results of Crawford et al.[41] that used the GEUVADIS dataset[5], from which 21 genes were identified in common (*AEBP1, CCDC126, CHI3L2, CSTB, ERAP1, FAM118A, FLVCR1-AS1, GJC1, LDHC, NMAT3, PEX6, PLA2G4C, POLR1E, RIBC2, SERPINB10, SLFN5, SYNGR1, TIMM10, TMEM106A, XRRA1, ZNF266*). The replication of a number of genes in our results across several independent studies demonstrates the utility of the approach, but the appearance of genes such as *LDHC* across multiple studies suggests these previous studies may also have been detecting signals linked to the same immortalisation pathways detected here rather than expression signatures of the original B cells. Consequently LCLs may provide a useful, readily accessible model of cellular proliferation, but their utility for mapping gene interactions relevant to primary cells may be more limited. Further application of this approach to large sets of tumour samples will be of particular interest to map how the observed genetic interactions overlap between different sets of immortalised cells.

We have presented a framework with which to identify putative genetic interactions that considers a gene's eQTL complement as opposed to performing an exhaustive search of individual eQTL. This approach substantially reduces the computational burden of a genome-wide exhaustive search, addressing to some extent the challenge of multiple testing. Our results indicate that interactions between distal genes involved in similar biological pathways potentially underlie the increased variance in expression associated with certain sets of regulatory variants for genes.

## Methods

**Ethics statement**. This investigation has been conducted in accordance with the Declaration of Helsinki and according to national and international guidelines. Ethics permission was obtained from the Multi-Centre Research Ethics Committee for Scotland (Wave 1: MREC/01/0/56), the Lothian Research Ethics Committee (Wave 1: LREC/2003/2/29), and the Scotland A Research Ethics Committee (Wave 2: 07/MRE00/58). All persons gave their informed consent prior to their inclusion in the study.

**Genotype data**. WGS data for 930 individuals of European ancestry from the LBC1936 cohort was generated on the Illumina X platform to a mean depth of 30X coverage and aligned to GRCh38 by Edinburgh Genomics (https://genomics.ed.ac.uk/). Variants were called using the HaplotypeCaller in GATK[44]. VCF files for each chromosome were filtered using vcftools[45] to retain variants with an 80% call rate, two observed alleles, a minor allele frequency (MAF) >0.01, minor allele count (MAC) >1, and minimum genotype quality (minGQ) of 40, and output in Plink[46] format. Data for 358 Europeans was sourced from the 1000 genomes project (GEUVADIS; ftp.1000genomes.ebi.ac.uk/vol1/ftp/release/20130502) and genomic positions were updated from GRCh37 to GRCh38 using CrossMap[47] (Supplementary Fig. 7). The datasets were filtered to retain only SNPs common to both datasets. Each dataset was processed in Plink using the rel-cutoff feature to remove individuals with observed genomic relatedness >0.025, resulting in a final dataset of 876 LBC1936 and 344 GEUVADIS individuals, and 7,477,426 SNPs. Conversion to dosage format was performed using the convert_plink_to_dosage.py script that accompanies PrediXcan[8]. Principal components analysis (PCA) was performed on LD filtered datasets using Plink (Supplementary Fig. 8). Briefly, the VCF files were filtered to remove SNPs within 1 Mb of one-another and having r2 > 0.1 using the –indep-pairwise (1000 kb 50 0.1) function in Plink.

**Gene expression data**. Expression data were acquired for the 876 LBC1936 individuals[48]. In summary, peripheral blood mononuclear cells (PBMCs) from 1091 LBC1936 individuals were extracted from whole blood collected at mean age 70 years (S.D. = 0.6 years) at the Edinburgh Clinical Research Facility (ECRF) Genetics Core, Western General Hospital, Edinburgh. PBMCs underwent Epstein-Barr Transformation to generate LCLs at the European Collection of Cell Cultures, Pubic Health England, Porton Down. Frozen LCL pellets were returned to the ECRF Genetics Core, where RNA was extracted using a Qiagen miRNeasy kit, biotin-labelled, and genome-wide gene expression levels measured using Illumina's HumanHT-12 v4 Expression BeadChip which contains 47,231 probes; a control RNA sample was included on each array. Individuals with a signal-to-noise ratio <10 or with fewer than 9000 detected transcripts (*P* < 0.01) were excluded (*n* = 130), and only probes expressed in >20% of individuals were retained[48]. The microarray data was quantile normalised using the Bioconductor package lumi[49], after which probes with no detectable expression were removed, resulting in 38,764 probes. A mixed model was applied in R using the lme4 package[50] to regress-out

sex, age, population structure covariates and technical variation by fitting sex, age, and the first five eigenvectors resulting from PCA of the genotype data in Plink (Supplementary Fig. 8) as fixed effects, and sample plate and Sentrix array as random effects:

$$\mathrm{expr}_i^A = \beta_0 + \beta_d \mathrm{age}_i + \beta_s \mathrm{sex}_i + \beta_1 \mathrm{PC1}_i + \beta_2 \mathrm{PC2}_i + \beta_3 \mathrm{PC3}_i + \beta_4 \mathrm{PC4}_i + \beta_5 \mathrm{PC5}_i + \mu_{\mathrm{plate},k} + \mu_{\mathrm{sentrixArray},m} + e_i \quad (1)$$

To remove hidden confounders the residuals from this model were further processed using PEER[51] with default settings, assuming 50 hidden factors and 1000 iterations. Convergence was reached after 290 iterations, revealing 14 hidden factors to be relevant. The residuals from Equation 1 were re-run in PEER, this time assuming 14 hidden factors, and the residuals of this analysis were used in lieu of the observed expression values when training the prediction models. A summary of the effects of this data pre-processing on the resulting prediction models is provided in the supplement (Supplementary Table 3).

Normalised gene expression data for 344 European individuals from the GEUVADIS dataset was sourced from the transcriptome sequencing undertaken by Lappalainen et al.[5], available at the EBI (https://www.ebi.ac.uk/arrayexpress/files/E-GEUV-1/analysis_results/). Briefly, the GEUVADIS expression data was generated by sequencing mRNA extracted from LCLs, generating an average of 49.8 million reads per individual after quality control. Further filtering was applied by the GEUVADIS consortium to scale read counts by the total number of mapped reads per sample to the median (GEUVADIS dataset from EBI: GD660. GeneQuantRPKM.txt.gz). We filtered this data further, as per GEUVADIS, to remove genes with 0 counts in >50% samples. Metadata for the GEUVADIS samples used in our study was sourced from EMBL-EBI's ArrayExpress database (https://www.ebi.ac.uk/arrayexpress/experiments/E-GEUV-1/samples/), from which we identified the assay ID and processing laboratory. These technical variables were regressed out as random effects, in addition to sample sex and PCA eigenvectors, using lme4 as described above. We ran PEER on the GEUVADIS data, as above, which reached convergence for 50 hidden factors after 223 iterations, revealing 17 hidden factors to be relevant. PEER was re-run for 17 factors, converging after 210 iterations, and the residuals from this were used in lieu of observed expression values for the GEUVADIS dataset when fitting the prediction models. We further filtered the GEUVADIS derived expression data to retain only genes with consistent Ensembl gene IDs between release 67, on which the GEUVADIS expression data genes were annotated, and release 88 on which the LBC1936 expression data was annotated. For each dataset, genes whose observed or predicted expression exceeded a kurtosis threshold of 9 (Supplementary Fig. 9) were removed to prevent potentially misleading results arising from abnormally distributed data. The moments package for R was used to calculate kurtosis.

A potential issue with the LBC1936 microarray data is that some genes were represented by multiple probes. If these different probes for the same gene are largely redundant this has the potential to skew the *P*-value distribution and consequently the FDR values. Previous studies have collapsed probes for the same gene into one value, for example by taking the mean, first principal component or top association[52], however part of the reason for including multiple probes for a gene on the arrays was because of the presence of different isoforms whose expression levels was not well captured by one probe. We observed many pairs of probes for the same genes that showed high correlation (e.g., see Supplementary Fig. 10) and consequently collapsing them into one value would likely reduce our ability to detect associations. We therefore instead tested whether the presence of multiple probes was actually skewing the *P*-value distribution and consequently FDR statistics. To do this we recalculated the FDR values of Equation 3 having excluded genes with multiple probes. As shown in Supplementary Fig. 11 the FDR values of the remaining genes were perfectly correlated to those when calculated including the multi-probe genes. This suggests that the presence of multiple probes for genes was not skewing this analysis.

**Testing variant-set-dependent variance in gene expression**. Using the LBC1936 dataset we employed PredictDB[8], which fits an Elastic Net linear model, to predict cis-eQTL from the set of SNPs located between 1 Mb upstream of each gene's transcription start site and 1 Mb downstream of each gene's termination site. We used PrediXcan to train models of gene expression levels for genes reported by PredictDB as having significant prediction models, which are those for which the coefficient of determination false discovery rate (FDR) was <0.05. Prediction model $R^2$ values were plotted against their respective gene sizes to determine if there was a correlation between a gene's size and its predictive capacity. The same models trained on the LBC1936 were used to predict gene expression levels in the GEUVADIS dataset as a means of validating the efficacy of the prediction models in an independent dataset. Genes with a coefficient of determination ($R^2$) ≤0.1 between their predicted and observed expression levels were considered poor gene models and removed from downstream analyses. Cross-validation therefore required gene models to have $R^2 \geq 0.1$ in both the LBC1936 and GEUVADIS datasets. We tested for evidence of set-dependent variance in expression levels in the LBC1936 by first fitting the following general linear model (GLM) in R:

$$\mathrm{obs}_i^A = \beta_0 + \beta_a \mathrm{pred}_i^A + e_i \quad (2)$$

Where $\mathrm{obs}^A$ are the residuals of Equation 1 and $\mathrm{pred}^A$ is the predicted expression of gene$^A$ in individual i. To determine if the variance in gene$^A$'s expression was

correlated to its set of cis-eQTLs, the Spearman's correlation was then calculated between pred$^A$ and the square of the residuals from the above model (Equation 2), for all unique gene$^A$. The same model was applied to the GEUVADIS dataset. These squared residuals consequently approximately equate to the gene's absolute expression levels having accounted for the effects of its cis-eQTLs. The correlation $P$-values were converted to FDRs using $R$'s p.adjust function, and the $-\log_{10}$(FDR) values of all gene$^A$ were compared to those subsequently identified as being involved in significant gene–gene interactions.

**Is variance in expression linked to distal gene expression?**. Linking set-dependent variance in gene expression to distal genes was performed by fitting the residuals from Equation 1 in the following model:

$$\text{obs}_i^A = \beta_0 + \beta_a \text{pred}_i^A + \beta_b \text{obs}_i^B + \beta_{ab} \text{pred}_i^A \text{obs}_i^B + e_i \quad (3)$$

Where pred$^A$ is the predicted expression of gene$^A$ in individual i, and obs$^B$ is the residual of Equation 1 for each other gene in the genome. The list of gene$^A$ comprised all genes passing the kurtosis threshold and with good prediction models ($R^2 \geq 0.1$ in both datasets), while the vector of gene$^B$ comprised all genes passing the kurtosis threshold. The aim of this model was to test if interaction effects between the predicted expression of gene$^A$ (i.e., its set of regulatory variants) and the observed expression of gene$^B$ can explain some of the observed variation in gene$^A$'s expression between individuals. The coefficient sign and ANOVA F-test $P$-values for the interaction term were retained, and the latter converted to an FDR to account for multiple testing.

**Do genetic interactions exhibit a bias in directionality?**. To explore whether or not there was a significant difference in the sign of the coefficient of significant gene pairs compared to the null, the sign of the interaction term for corresponding gene pairs from the LBC1936 and GEUVADIS were summarised, providing a count of gene pairs for each sign combination across the two datasets: [−/−], [−/+], [+/−], [+/+]. A summary of counts was generated for all gene pairs tested in both datasets to represent the null distribution irrespective of significance. A chi-squared test was performed in $R$ to determine if there was a significant difference in the distribution of coefficient directionality between the null and significant gene pairs.

**Comparing regulatory variant-sets to individual cis-eQTL?**. To evaluate how the variant-set approach performed when compared to the single best eQTL for predicting gene expression and gene–gene interactions, FastQTL[53] was run on the 4122 genes (5,033 gene probes) that passed the kurtosis filters in both datasets. The VCF files provided to FastQTL contained the same SNPs analysed by PredictDB in the variant-set analyses, together with the same expression data after having regressed out covariates and performing PEER normalisation. The PCA eigenvectors used as covariates in the modelling for the variant-set analyses were also provided as covariates to FastQTL. The programme was run with seed 123456789 for 1,000 permutations using the default cis-window size (1 Mb—consistent with the PredictDB analyses). For each gene, FastQTL identifies the SNP best associated with the expression data (Supplementary Data 13 and 14). The allele dosage data for each of the identified SNPs was then regressed against the observed expression for its gene, generating $R^2$ values in the same manner as the variant-set analyses. Genes whose eQTL resulted in good prediction of expression values ($R^2 > 0.1$) were retained. These genes were subject to the same gene–gene interaction modelling as the variant-set analyses (Equation 3) but with predicted gene expression pred$^A$ replaced with allele dosage data for the SNP identified by FastQTL. The significant concordant interactions identified from this analysis are provided in Supplementary Data 15.

**Can genetic interactions be explained by an intermediary?**. Of the gene pairs that returned significant results (ANOVA F-test FDR <0.05), the model was extended further to test (1) if the significant gene–gene interaction could be explained by any other genetic variants proximal to gene$^A$:

$$\text{obs}_i^A = \beta_0 + \beta_a \text{pred}_i^A + \beta_b \text{obs}_i^B + \beta_{ab} \text{pred}_i^A \text{obs}_i^B + \beta_c \text{snp}_i^A + e_i \quad (4)$$

Where snp$^A$ represents any SNP within the entire range from 1 MB upstream of the transcription start site of gene$^A$ to 1 MB downstream of its termination site; or (2) if the interaction could be explained by the expression of a third gene:

$$\text{obs}_i^A = \beta_0 + \beta_a \text{pred}_i^A + \beta_b \text{obs}_i^B + \beta_{ab} \text{pred}_i^A \text{obs}_i^B + \beta_c \text{obs}_i^C + e_i \quad (5)$$

Where obs$^C$ is the residual from Equation 1 of any other gene passing the kurtosis filter. If the interaction term explained no more variation in gene$^A$'s expression than nearby variants or other genes then it was assumed that a simpler explanation was that the observed interaction was in fact driven by one of these secondary factors.

**Is the genetic component the driver of genetic interactions?**. To test for the evidence of genetic interactions underlying the gene–gene interactions, we fitted an interaction between the predicted expressions of both genes, i.e., their cis-genetic

effects, as described below:

$$\text{obs}_i^A = \beta_0 + \beta_a \text{pred}_i^A + \beta_b \text{pred}_i^B + \beta_{ab} \text{pred}_i^A \text{pred}_i^B + e_i \quad (6)$$

As an alternative approach we also tested which interactions remained after accounting for each of the individual eQTLs associated with gene$^B$ in the PrediXcan model. To do this we extended the model that tested for an interaction between obs$^A$ and pred$^A$ to account for the genotypes of all of the cis-regulatory variants identified by PrediXcan for gene$^B$:

$$\text{obs}_i^A = \beta_0 + \beta_a \text{pred}_i^A + \beta_b \text{obs}_i^B + \beta_{ab} \text{pred}_i^A \text{obs}_i^B + \beta_{s,1} \text{snp}_i^{B,1} \ldots + \beta_{s,n} \text{snp}_i^{B,n}$$
$$+ \beta_{as,1} \text{pred}_i^A \text{snp}_i^{B,1} \ldots + \beta_{as,n} \text{pred}_i^A \text{snp}_i^{B,n} + e_i \quad (7)$$

where snp$^B$ represents any eQTL between 1 MB upstream of the transcription start site and 1 MB downstream of the termination site of gene$^B$. Although there was the potential for overfitting in this analysis on average there were 817.36 ± 35.38 S.D. and 293.62 ± 31.21 residual degrees of freedom in the LBC1936 and GEUVADIS cohorts respectively.

**Does gene co-expression explain the genetic interactions?**. To investigate the relationship between these genetic interactions and gene co-expression, we applied the Pearson's correlation statistic to the observed expression of genes in significant pairs with concordant coefficient signs in both datasets. These tests were performed in $R$ and $P$-values converted to FDRs. To determine if there was a linear relationship between datasets with regards to the co-expression of genes, the $R^2$ values resulting from the previous test for the two datasets were compared using the Pearson's correlation statistic.

**Enrichment analyses**. FUMA[54] was used to test for the enrichment of particular biological pathways and motifs among the set of interacting gene$^B$s of each gene$^A$ using the hypergeometric test. In this analysis the background gene list was all gene$^B$s tested (irrespective of significance) and consequently only included genes expressed in greater than 50% of the samples. The combined set of all, non-redundant interacting gene$^B$s were also tested in the same way. To investigate the binding of EBV latency proteins and NF-κB subunits at the promoters of the interacting gene$^B$s, ChIP-seq data processed by Jiang et al.[16] was obtained from the WashU epigenome browser[55] (http://epigenomegateway.wustl.edu/browser/?genome=hg19&session=AuL8qiK9Bf). To investigate if the observed binding of these proteins at the promoters of interacting gene$^B$s was greater than expected by chance, the same number of tested (but not necessarily significant) gene$^B$s were randomly selected from the background list 100 times. These were used to calculate a 95% confidence interval around the median binding levels observed across the total background list of gene$^B$s (Supplementary Table 2).

**Variance explained by additive and interaction effects**. To determine the amount of gene expression variance explained by the identified interactions the following models were employed:
cis-genetic

$$\text{obs}_i^A = \beta_0 + \beta_a \text{pred}_i^A + e_i \quad (8)$$

cis-genetic+distal gene main effects

$$\text{obs}_i^A = \beta_0 + \beta_a \text{pred}_i^A + \beta_b \text{obs}_i^B + e_i \quad (9)$$

cis-genetic+distal main+distal interaction

$$\text{obs}_i^A = \beta_0 + \beta_a \text{pred}_i^A + \beta_b \text{obs}_i^B + \beta_{ab} \text{pred}_i^A \text{obs}_i^B + e_i \quad (10)$$

Models were compared with ANOVA and the additional variance explained by additive and interaction effects was then determined from pseudo $R^2$ values calculated by subtracting the residual deviance from the null deviance and dividing the result by the null deviance.

The sum of the additional variation in a gene's expression explained by its interactions was calculated by performing stepwise regression. Due to the large number of interactions involving some gene$^A$s, to address the risk of overfitting, for each gene$^A$ we removed gene$^B$s whose observed expression levels were correlated to others. This process was performed in $R$ using the findCorrelation function of the caret package[56] with an absolute correlation cutoff of 0.4. For all gene$^A$ with ≥2 interactions the above cis-genetic, additive and interaction models were extended to include all gene$^B$ for a given gene$^A$ after removing this redundancy. Stepwise regression was performed 100 times, and for each iteration the order of gene$^B$ was randomised and the data split into an 80% training and 20% test set. The mode of stepwise search included both forward and backward directions, with the lower scope being the fit of the cis-genetic model and the upper scope being the full interaction model. Following each iteration, the deviance, predictors and Akaike information criterion (AIC) were recorded from the model on which regression converged. The results were summarised in $R$. For each set of predictors the model was compared to the residual by ANOVA using the F-test. Pseudo $R^2$ values were calculated from the test set for each model with and without the interaction terms to determine the amount of extra variance explained by the interactions.

**Reporting summary**. Further information on research design is available in the Nature Research Reporting Summary linked to this article.

## Data availability

Genotype and normalised gene expression data for the GEUVADIS dataset is available at: ftp.1000genomes.ebi.ac.uk/vol1/ftp/release/20130502 and https://www.ebi.ac.uk/arrayexpress/files/E-GEUV-1/analysis_results/, respectively. Sequence data for the Lothian Birth Cohort has been deposited at the European Genome-phenome Archive (EGA), which is hosted by the EBI and the CRG, under accession numbers EGAS00001003818 and EGAS00001003819.

## Code availability

Statistical modelling was performed in R v3.5.1, all tests are two-tailed with the exception of the one-tailed hypergeometric test of FUMA. Computer code is available upon request.

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

## Acknowledgements

J.G.D.P. and D.W. are supported by the Biotechnology Sciences Research Council (BBSRC Grant No. BBS/E/D/10002071). The LBC1936 is principally supported by Age UK (Disconnected Mind grant). The LBC1936's whole-genome sequencing was funded by the BBSRC. The LBC1936's lymphoblastoid cell lines' gene expression resource was funded by the University of Edinburgh's College of Arts, Humanities and Social Sciences. I.J.D. and S.E.H. are supported by the Centre for Cognitive Ageing and Cognitive Epidemiology, which is funded by the Medical Research Council and the BBSRC (MR/K026992/1). We thank the LBC1936 participants and LBC1936 research team for their contributions, and in particular Dave Liewald for his assistance.

## Author contributions

J.P. conceived the study and J.P., D.W., H.B. and A.T. designed the analyses. S.E.H., D.H. and I.J.D collected samples and generated RNA and DNA data. A.J.B. conducted the DNA variant calling. V.R. and C.P. conducted RNA microarray data processing. D.W., Q.L. and Z.L. conducted modelling. D.W. and J.P. conducted statistical analyses and prepared the initial manuscript. All authors contributed and commented on the development of the manuscript.

## Competing interests

The authors declare no competing interests.
