## [Peer Review File · Nature Communications]

Reviewers' comments:

Reviewer #1 (Remarks to the Author):

1) What are the major claims of the paper?

This paper describes the how cis-eQTLs can be combined into regulatory haplotypes and how these haplotypes can be used to determine differential expression of distal genes.

These distal genes are then fitted into gene-gene interaction networks to implicate important cellular pathways.

2) Are they novel and will they be of interest to others in the community and the wider field?

I think this paper would have a wider audience if the work could be presented in the context of understanding disease mechanism. Could the authors comment on how their approach could be useful in seeking to understand expression differences in specific diseases and the relevance of risk alleles for those diseases?

3) The conclusions of the paper are novel

4) Is the work convincing, and if not, what further evidence would be required to strengthen the conclusions?

I wasn't entirely convinced about the approach being used. I can appreciate that using predicted expression data is a useful tool to allow incorporation of some of the older microarray datasets in this type of analysis, but I would have expected that the authors would use RNA-Seq datasets (test and replication) as the primary source for their analyses and then to use predicted microarray data for validation.

5) On a more subjective note, do you feel that the paper will influence thinking in the field?

If the manuscript is revised to more fully integrate the statistical analysis with the biological interpretation, then the impact of the paper will be greater.

6) Please feel free to raise any further questions and concerns about the paper.

What was the ancestry of the individuals used for this project? If not of European ancestry, can the authors comment on the effect that ancestry may have on their conclusions given the differences in LD structure between European and non-European ancestries.

Was any attempt made to identify candidate transcription factors binding to the regulatory haplotypes?

The biological relevance of the findings is somewhat drowned in the statistical analyses, so the significance of the paper is somewhat lost.

The paper needs to be restructured so that at each stage of analysis, the biological relevance of the results are made clear.

7) We would also be grateful if you could comment on the appropriateness and validity of any statistical analysis, as well the ability of a researcher to reproduce the work, given the level of detail provided.

The statistical methods seem appropriate and sufficiently detailed for a statistician to undertake

Reviewer #2 (Remarks to the Author):

Wragg et al. presented an interesting new framework for detection of gene-gene interactions (epistasis) in transcriptome data with much reduced burden of multiple testing. Unlike the previous veQTL approach focusing on eQTLs associated with substantial variability of gene expression levels, the authors concentrated on genes with expressions driven mainly by cis-eQTLs and generated so called 'regulatory haplotypes' per gene by aggregating the member cis-eQTLs and then tested the interactions of the haplotypes with expression levels other genes in the genome. Using this framework, the authors identified a set of haplotype epistatic interactions that are statistically replicated and biologically plausible.

I hope the following comments are helpful:

A. While the idea is pretty exciting, this paper seemed to sell two stories (i.e. a novel approach and a real-life application) in one go, making a little bit difficult to follow sometimes. The method may be stronger if the following issues could be addressed:

- lines 71-73: the construct seems to imply an assumption of additive mode among cis-variants, which however could be either up- or down-regulated. Would the method risk missing interacting genes if cancelling events exist?
- Do gene sizes matter in the selection of genes/haplotypes, e.g. a bigger gene likely regulated by more cis-eQTLs? Boxplots of the R² values and gene sizes of the 744 (1st selected from LBC1936), 258 (overlapped in 1kg) and 59 (with haplotype-dependent variance) genes might be useful to answer these questions.
- While the haplotype filtering is useful to increase power of detection of epistasis, how to control false positives could be a major concern, particularly when only FDR threshold was used. The authors managed the issue by applying statistical replication in an independent cohort (likely stringent) where sample size is a lot smaller and samples could be a lot younger. However, a good discussion of the issue would be very useful for potential users of your framework.
- Discussion of other pros and cons of the framework would also be useful, including those that the authors have done well in examining differences among veQTL, haplotype-dependent veQTL, epistatic haplotypes, directions of interactions, inter-chromosome interactions. Hints to supplement functional evidence of epistatic eQTLs would also help to make the framework more useful.

B. My major concern on the biology side is the impact of old age (70 ± 0.6) on gene expressions in the LBC1936 cohort, although the authors have adjusted for the age effects. It is interesting to know whether the identified interaction networks have some relevance with aging, and whether the method might be more powerful if it were applied in a young cohort. Some discussion regarding the aging factor would be nice, given the 1KG cohort samples are very unlikely as old as 70.

C. In the Methods section

- equations should be numbered
- lines 426-427: 'vector' appeared the first time - do you really mean it?
- line 446: don't understand 'any SNP within 1MB upstream and 1MB downstream' - do you fit every SNP within the ranges or SNPs after pruning for LD? fitting orders matter?
- line 478: how did you control overfitting here?
- lines 442 to 501 read difficult to me

Minor points:

- Title in page 1: not sure if 'defining' is the right word here. Perhaps 'detecting'?
- lines 37-38: perhaps to include 'gene-environment interactions'
- line 45: which one 'unprecedented'?

- line 76: Figure 1 should appear and be explained in the first part of the results
- line 143: 'higher' in what sense?
- line 149: the subheading 'Haplotype-dependent variance in expression' is confusing because you tested 258 genes of which only 41% with haplotype-dependent variance in expression; only a proportion of the claimed epistatic interactions involved genes within the 41%
- lines 214-215: not sure if it is appropriate to test against randomness here as the 59 genes were highly selected. It is perhaps better to question here why the other ~50% did not interact, due to low power or no replication or something else?
- lines 264-267: this is an important point saying the advantage of the framework, so better to be further discussed and highlighted in e.g. Discussion
- lines 300-301: why would be the case given your samples are likely healthy in both cohorts?
- line 318: give reference for "Zhernakova et al."
- line 581-582: P values were zero in Figures 5c and 5d, typo?

Wen-Hua Wei

Reviewer #3 (Remarks to the Author):

Manuscript number: NCOMMS-18-7184855

Title: A novel approach for defining gene-gene interactions identifies networks of genes linked to cell immortalisation.

In this manuscript Wragg et al, perform a gene-gene interaction (GxG) analysis in LCLs using the genetic contribution of gene expression as an aggregate of the cis-variants in haplotypes instead of the gene expression measurements. The authors also performed a veQTL analysis, since genes exhibiting variance effect are influence by either GxG interactions or genotype-by-environments (GxE) interactions. By this, the authors further tested interactions only for the genes with significant veQTLs effects. This approach allows reducing the multiple testing burden of standard GxG analysis with a large searching space while it aims to capture the total genetic contribution of an individual genotype to gene expression. As a result, the authors identify GxG interactions in expression from a microarray study (LBC1936), replicating many of their finding in the GEUVADIS dataset.

The manuscript is very well written. The authors manage to explain complex concepts in a way that is easy to understand and their work is relatively easy to follow. I am positive about the work and consider that it is an interesting addition to the recent studies using similar approaches to identify non-additive genetic effects. Their solution is however not completely novel, but I believe it is important to show how the combination of genetic and variance effects can be used to identify trans associations. Please note that the use of predicted expression from the genetic component to identify GxG interactions has been presented by the authors of PrediXcan in a recent preprint (Wheeler et al, BioRxiv 471748).

There are number of issues, one of which I believe may influence the results. In addition, some of the conclusions presented in the discussion are poorly supported by the results and need some clarifications:

1) Regarding the LBC1936 dataset, the authors need to explain how they treated the difference probes during their multiple testing corrections. Multiple probes report expression levels for the same gene, giving highly correlated phenotypes. Those correlations influence the distribution of the pvalues and therefore the multiple testing corrections. Hopefully, an adequate multiple testing correction will not modify much the list of genes included giving the authors are only testing 258 genes for gene-gene interactions, but an appropriate correction is needed here and it will influence the list of genes provided as a result.

2) Most of the discussion, and the title of the manuscript, make reference to the network associated to LDHC. From a network of GxG interactions, the authors link their work to the LCLs transformation process and to cancer. This is completely speculative and not supported by the results. Are the authors implying causality here? Because of this, I find the title misleading. They

have no evidence for the implication of the GxG interaction network to cell immortalisation, only of the relationship between the genes and the genetic component of LDHC. The fact that many of those genes have been implicated in immortalization is a supporting point for the hypothesis that the network itself is involved, but at this point, and as presented in the manuscript, this is only a hypothesis that the authors are not actually testing any further. On the other hand, the two larger networks have in common their relationship with metabolism, and the GO enrichment analysis showing "ATP synthesis coupled electron transport" and "Oxidative phosphorylation" seem to support this more strongly than the cancer associations the authors are presenting. Unless the authors can provide stronger evidence for the link of the network with cell immortalisation, these strong statements need a revision.

3) The correlation analysis is slightly confusing. First, the authors need to explain why is it relevant to perform a correlation analysis between observed expressions, while the main advantage of the approach presented here for gene-gene interactions is using predicted expression. What is correlation when using the same phenotypes used in the interaction analyses?

Second, the main conclusion presented as a consequence of the analysis is "that restricting the analysis to just co-expressed genes as in previous studies would have potentially missed a large number of the putative interactions identified in the current study.". This need a citation (Is it maybe Zhernakova et al, 2017?) and an explanation, since it is the first time the authors are suggesting it is common practice to subset genes for gene-gene interactions analyses using correlations. In addition, this argument is contradictory with the main premise of the manuscript. If testing only correlated genes misses a large number of putative interactions, then testing only genes with veQTLs, as the authors have done here, have also the potential to miss a large number of putative interactions. The authors seem to criticize an approach similar to theirs without admitting their own limitations in the matter.

4) Please cite the R packages or any other software used. For example, what package was used for the implementation of the GLM models?

Minor comments:

- Page 10, line 258 = write range of values for variance explained.
- Page 7, line 180: I believe the authors mean "testis" instead of "testes".
- In the manuscript, the RNAseq dataset is named "1kG cohort", "1kG dataset", or even the more common consortium name that produced the data "GEUVADIS". The authors need to choose one name and be consistent through the manuscript to avoid confusing the reader.

Reviewer #4 (Remarks to the Author):

The manuscript "A novel approach for defining gene-gene interactions identifies networks of genes linked to cell immortalization" by Wragg et al., describes a new framework for testing for non-additive interactions between a collective set of genetic determinants of cis regulatory effects on expression of a given gene and the expression levels of distal genes, such as transcription factors. This approach tests the interaction of a set of regulatory variants as a single component (what they term cis-regulatory haplotypes) identified using predictive models, such as PrediXcan, with the expression levels of other genes in the genome, as opposed testing each regulatory variant (eQTL) individually. This enables to decrease the multiple hypothesis burden and to potentially increase power of detection. In addition to providing new biological insight into the molecular mechanisms of gene regulation, the authors show that such interactions can explain a substantial amount of the unexplained variation of gene expression differences between individuals (4-10%) for a subset of eQTLs. The authors test their framework on 876 lymphoblastoid cell lines (LCLs) from a Scottish birth cohort, and replicate their results in an independent set of LCL samples from European individual from the 1000 Genomes project. Using a general linear regression model, the authors identify 38 genes out of an initial set of about 700 genes with significant interactions between the cis genetic determinants of their expression levels with the expression of distal genes. They show that these genes are involved in ATP synthesis that enables immortalized cells to

rapidly proliferate.

This is the first time to that interactions between a set of variants controlling gene expression and the expression of another gene/trans factor are systematically tested. While the approach and methodologies used are solid and the results are of broad interest, I have several comments/concerns about the normalization and covariate correction applied to the expression data, and the evaluation of this approach compared to the typical approach of testing interactions between individual variant eQTLs versus a set of regulatory effect with the expression of other genes. I also provide a few suggestions for clearer presentation of the results.

Major comments:

1. From the analyses in the paper, it is not clear to what extent using the regulatory haplotypes that collectively explain a larger proportion of expression variation of a given gene, increases power of detecting non-additive gene-gene interactions relative to individual genetic regulatory effects on expression. Can the authors compare their non-additive interaction results using regulatory haplotypes to that using individual variant effects (eVariants), e.g., using the most significant eQTL per gene_A or testing individually each of the independent eQTLs per gene? Testing for interactions with individual eQTLs might be advantageous when there are several haplotypes with different or opposite effects on the expression levels of a given gene and only one of the haplotypes interacts with a distal gene.

2. The authors use the expression 'regulatory haplotypes' to represent a set of variants that collectively contribute to the gene expression variation. I feel that this terminology should be changed as it is not precise. Haplotypes refer to a set of alleles that lie on the same chromosome, usually identified through statistical phasing of genotype data or long read sequencing. The regulatory haplotypes referred to in the paper are a set of variants whose genotypes are associated with gene expression levels, however, they may be on separate haplotypes. I would recommend using a different term, such as 'regulatory variant sets'.

3. On page 14, in the methods section, quantile normalization is applied to the array data to align the distributions of all probes/genes between samples. It is also important however that the probe or gene expression levels are normally distributed across samples for the performance of the linear regression model employed in PrediXcan and possibly for the general linear models used to test for non-additive interactions. The authors mention this in the Discussion on page 11, line 275, and state that they attempt to address this through their replication scheme. Have the authors checked the distribution of expression of each gene across samples? A common normalization approach to ensure each gene's expression levels are normally distributed across samples is standard normalization (e.g., as done in GTEx, PMID: 29022597) or log transformation. I would recommend the authors check how applying standard normalization to the expression data from the birth cohort affects their results. This is also expected to reduce the effect of outliers. Also, on page 14, can the authors briefly mention how the gene expression data for the 344 European individuals from the 1kG dataset were normalized.

4. In the general linear model of the observed gene expression of a given gene (in Methods, page 15), the authors adjusted for sex, age, and population stratification using the first five genotype principal components (PCs). However, they do not adjust for potential batch effects or hidden experimental or other confounding factors on gene expression variation, which have been shown in many studies to exist. The authors note at the end of the Results section on the bottom of Page 10, that some of the discrepancies between the two studies, the birth cohort and 1KG, could be due to technical variation in gene expression measurements in the array study vs. the RNA-seq study. Did the authors choose not adjust for hidden covariates in expression data as they were concerned that this would remove true trans effects on expression variation? I would suggest testing how the number of non-additive interaction changes with adding in hidden expression covariates, for example using the Probabilistic Estimation of Expression Residuals (PEER) method (PMID: 22343431), as applied for example for the trans eQTL analysis in GTEx (PMID: 29022597).

5. The authors argue in the Introduction and in their simulated example in Figure 3 that detecting

variance eQTLs suggests that there may exist non-additive interactions between the eQTL and the expression levels of a transcription factor. However, the authors do not investigate this hypothesis further in the interactions they find between the regulatory haplotypes and the expression of distal genes. For example, are the distal genes found to interact with the genetically determined expression levels of a given gene enriched for known transcription factors or other regulators?

Furthermore, on page 7 in the Results section, the authors note that a large proportion of the 244 high confidence interactions found are centered on a common set of genes. Based on the hypothesis that transcription factors may underlie much of the non-additive interactions, I would have expected to see an opposite pattern, i.e. interactions between a single trans regulator (distal gene_B) and multiple target genes (gene_A). Do any of the interactions show such a pattern? Why are we not seeing this pattern as the more common pattern? Also, can the authors note how many target genes, i.e. 'gene_A' comprise these 244 interactions and show the distribution of number of interactions per gene_A.

6. To make it easier to follow all the results in the paper I would recommend summarizing the main findings in a table, e.g., number of genes with a significant PrediXcan models, number of significant veQTLs, number of significant GxG interactions, intersections, etc., with a column for significance in the Scottish birth cohort, 1KG and in both.

7. Such interactions between distal genes and the genetic determinant of gene expression of another gene were found for only 20% or less of genes tested. Can the authors comment on possible reasons this may be.

Minor comments:

1. In Figure 1b, the model of the observed expression levels of a given gene is shown only as a function of an interaction term between the genetically determined component (in cis) of gene expression of this gene and the observed expression of another distal gene, but it should also include these effects individually and other potential effects (e.g., environmental effects not related to the distal gene's expression levels).

2. In the introduction, on page 3, can the authors add a reference to support the sentence on lines 49-50: "Each cis-eQTL has, though, been found to generally only explain a small proportion of the variation in a gene's expression between individuals.", e.g., Gamazon, E. R. et al. A gene-based association method for mapping traits using reference transcriptome data. Nat. Genet. 47, 1091–1098 (2015).

3. On page 5, lines 116-117, can the authors add a reference to the sentence: "Previous studies have highlighted how both genetic and environmental interactions can lead to genotype dependent variance in a gene's expression."

4. On page 7, in the sentence on lines 178-179, for further clarification I might add the word 'trans' to the 'untested factor' (i.e. untested trans factor driving the pathway).

5. On page 8, line 194, is the R^2 a Pearson or Spearman correlation of determination?

6. In the methods section on page 15, it says that only genes reported by PredictDB as having a significant prediction model were trained with the PrediXcan model in the Lothian birth cohorts. Why did the authors not test all genes, since the model significance should be dependent on the study in which it was trained.

7. In the methods section on the bottom of page 17, can the authors briefly state what statistical test is used in ConsensusPathDB to assess gene set enrichment in the given subnetworks.

8. I would recommend increasing the font of the genes in Figure 6, at least those in the hubs.

9. In figure 8, are the units of the y-axis in panels A, B, and C percentage or fraction? Based on the results I assume it is percentage?

RESPONSE TO REVIEWERS

We would like to thank the reviewers and editors for their insightful comments that we firmly believe have substantially improved the clarity and quality of the analyses. This has included, but is not restricted to:

- Reanalysing the results after using PEER which has led to an increase in the number of interactions identified.
- Comparing how results change if only the single best eQTL for each gene is used instead of the regulatory variant set.
- Improving the interpretation of the links to cancer by building on the previously reported observations with further enrichment analyses and using ChIP-seq data to investigate the binding of transcription factors that are linked to immortalisation at the promoters of the interacting genes.

For each point raised we highlight in more detail below how and where we have addressed it in the updated manuscript.

Many thanks

Dr James Prendergast

Reviewer #1 (Remarks to the Author):

1) What are the major claims of the paper?

This paper describes the how cis-eQTLs can be combined into regulatory haplotypes and how these haplotypes can be used to determine differential expression of distal genes.

These distal genes are then fitted into gene-gene interaction networks to implicate important cellular pathways.

2) Are they novel and will they be of interest to others in the community and the wider field?

I think this paper would have a wider audience if the work could be presented in the context of understanding disease mechanism. Could the authors comment on how their approach could be useful in seeking to understand expression differences in specific diseases and the relevance of risk alleles for those diseases?

We have added further analyses that reinforce the observed links between the identified gene networks and cancers. As well as new enrichment analyses which strongly implicate cancer gene networks, for example those linked to nasopharyngeal carcinomas and the MYCN proto-oncogene, we have added a new analysis of transcription factor binding that illustrates that the interacting genes in these networks are preferentially bound by key EBV TFs and NF- κ B subunits, associated with driving transformation (see lines 267 to 298). Consequently we illustrate how this approach can provide mechanistic insights of different phenotypes and we go into more detail on this in the response to point 2 of reviewer 3 below.

3) The conclusions of the paper are novel

4) Is the work convincing, and if not, what further evidence would be required to strengthen the conclusions?

I wasn't entirely convinced about the approach being used. I can appreciate that using predicted expression data is a useful tool to allow incorporation of some of the older microarray datasets in this type of analysis, but I would have expected that the authors would use RNA-Seq datasets (test and replication) as the primary source for their analyses and then to use predicted microarray data for validation.

We do strongly believe that the fact that the interactions we observe replicate across not only populations but also platforms is a strength rather than a weakness. Previous studies have for example illustrated how normalisation procedures can increase false positives (Castaldi et al, PloS One 2017). As the data from the different technologies were processed differently this should help minimise this effect. The microarray dataset has other advantages, for example that it is an unusually large cohort of individuals all of the same age from the same, relatively small geographic area, reducing these potential confounders (we now highlight this at line 334). Ultimately our results show that this approach does not appear to depend on using the same expression assay platform across cohorts, increasing its usability.

5) On a more subjective note, do you feel that the paper will influence thinking in the field?

If the manuscript is revised to more fully integrate the statistical analysis with the biological interpretation, then the impact of the paper will be greater.

Many thanks. As we note in the response above and further in response to reviewer 3 we have now expanded the biological analyses. See for example pages 10-11 in results section and 13-14 in the discussion.

6) Please feel free to raise any further questions and concerns about the paper.

What was the ancestry of the individuals used for this project? If not of European ancestry, can the authors comment on the effect that ancestry may have on their conclusions given the differences in LD structure between European and non-European ancestries.

Yes, all individuals in both cohorts were of European ancestry (see lines 107 and 119, and 427 and 432 in results and methods sections)

Was any attempt made to identify candidate transcription factors binding to the regulatory haplotypes?

We have now added motif enrichment and TF binding analyses at lines 291 to 298.

The biological relevance of the findings is somewhat drowned in the statistical analyses, so the significance of the paper is somewhat lost.

The paper needs to be restructured so that at each stage of analysis, the biological relevance of the results are made clear.

We have edited the manuscript to further emphasise the biological importance of the analysis along these lines. For example see lines 266-290 and 364-373.

7) We would also be grateful if you could comment on the appropriateness and validity of any statistical analysis, as well the ability of a researcher to reproduce the work, given the level of detail provided.

The statistical methods seem appropriate and sufficiently detailed for a statistician to undertake

Reviewer #2 (Remarks to the Author):

Wragg et al. presented an interesting new framework for detection of gene-gene interactions (epistasis) in transcriptome data with much reduced burden of multiple testing. Unlike the previous veQTL approach focusing on eQTLs associated with substantial variability of gene expression levels, the authors concentrated on genes with expressions driven mainly by cis-eQTLs and generated so called 'regulatory haplotypes' per gene by aggregating the member cis-eQTLs and then tested the interactions of the haplotypes with expression levels other genes in the genome. Using this framework, the authors identified a set of haplotype epistatic interactions that are statistically replicated and biologically plausible.

I hope the following comments are helpful:

A. While the idea is pretty exciting, this paper seemed to sell two stories (i.e. a novel approach and a real-life application) in one go, making a little bit difficult to follow sometimes. The method may be stronger if the following issues could be addressed:

- lines 71-73: the construct seems to imply an assumption of additive mode among cis-variants, which however could be either up- or down-regulated. Would the method risk missing interacting genes if cancelling events exist?

We believe this comment is along the same lines as comment 1 of reviewer 4 i.e. integrating the effects of multiple cis-regulatory variants may be masking interactions that for example only involve one of the cis-regulatory variants. To explore this we repeated the analysis but this time only taking the single best eQTL for each gene as suggested by reviewer 4. As shown in Supplementary Figure SF2, 36.1% of interactions were identified only when using the full set of cis-regulatory variants, supporting the utility of this approach. Both reviewers are likely right that some interactions are potentially masked using the full set of cis-regulatory variants as a subset of interactions were only identified when using just the single best eQTL. We now present these results at lines 194 to 207 and provide the set of interactions identified using the single best eQTL as Supplementary Table ST17. Many thanks for the suggestion.

- Do gene sizes matter in the selection of genes/haplotypes, e.g. a bigger gene likely regulated by

more cis-eQTLs? Boxplots of the R² values and gene sizes of the 744 (1st selected from LBC1936), 258 (overlapped in 1kg) and 59 (with haplotype-dependent variance) genes might be useful to answer these questions.

The R² of the prediction model was observed to be largely uncorrelated to the gene's size (R² of 0.0017; Supplementary Figure SF1). We now highlight this at line 111.

- While the haplotype filtering is useful to increase power of detection of epistasis, how to control false positives could be a major concern, particularly when only FDR threshold was used. The authors managed the issue by applying statistical replication in an independent cohort (likely stringent) where sample size is a lot smaller and samples could be a lot younger. However, a good discussion of the issue would be very useful for potential users of your framework.

Statistical replication of the interactions identified in the elderly LBC1936 cohort in the GEUVADIS dataset, which does not exhibit the same age bias, using data generated with different platforms, we feel, significantly reduces the likelihood of false positives. We have emphasised this in the discussion at lines 323 and 334.

- Discussion of other pros and cons of the framework would also be useful, including those that the authors have done well in examining differences among veQTL, haplotype-dependent veQTL, epistatic haplotypes, directions of interactions, inter-chromosome interactions. Hints to supplement functional evidence of epistatic eQTLs would also help to make the framework more useful.

We have now included further commentary in the discussion at lines 382-393 to highlight the advantages and limits of the framework.

B. My major concern on the biology side is the impact of old age (70 ± 0.6) on gene expressions in the LBC1936 cohort, although the authors have adjusted for the age effects. It is interesting to know whether the identified interaction networks have some relevance with aging, and whether the method might be more powerful if it were applied in a young cohort. Some discussion regarding the aging factor would be nice, given the 1KG cohort samples are very unlikely as old as 70.

The reviewer is correct that the LBC data is derived from a collection of elderly individuals of largely the same age at sampling. However there is actually unusually little variation in age among these individuals (as were all from the same year group at school) meaning the variation in expression between individuals is probably less likely due to differences in their ages. As we required any interactions to replicate in the younger GEUVADIS cohort we believe any associations that are specific to this unusually elderly cohort are unlikely to replicate. We have added a discussion of this at line 334. Reassuringly the enriched terms in the new FUMA analysis do not show evidence of being specifically linked to age with much stronger links to immortalisation pathways.

C. In the Methods section

- equations should be numbered

Equations are now numbered and have been re-written for consistency.

- lines 426-427: 'vector' appeared the first time - do you really mean it?

Thanks, we have now changed this to “list” (see line 537).

- line 446: don't understand 'any SNP within 1MB upstream and 1MB downstream' - do you fit every SNP within the ranges or SNPs after pruning for LD? fitting orders matter?

Yes this follows the standard PredictDB pipeline where all variants between 1MB upstream of the TSS and 1MB downstream of the TES are fitted. We have rephrased this earlier in the text to try and make this clearer (see line 576).

- line 478: how did you control overfitting here?

On average only 25 eQTLs were fitted in each of these models with a maximum of 137 variants. So the number of variables being fitted was quite a lot lower than the number of individuals/degrees of freedom minimising the risk of over-fitting. We have now made this clear at line 601. For overfitting to be an issue in this analysis it would need to lead to the loss of the significance of the interaction, but in fact this only happened for five gene pairs across both datasets, suggesting that the gene B eQTLs largely could not explain the observed interaction.

- lines 442 to 501 read difficult to me

We have tried to clarify this section (now lines 570-652).

Minor points:

- Title in page 1: not sure if 'defining' is the right word here. Perhaps 'detecting'?

We have changed ‘defining’ to ‘detecting’.

- lines 37-38: perhaps to include 'gene-environment interactions'

Thanks, we have now changed the start of the following sentence to “Understanding how these factors combine and potentially interact to shape a gene’s expression” (line 40).

- line 45: which one 'unprecedented'?

We have now changed ‘unprecedented’ to ‘substantial’ (line 48).

- line 76: Figure 1 should appear and be explained in the first part of the results

Figure 1 has now been cited in the results rather than the introduction.

- line 143: 'higher' in what sense?

We have now tweaked this sentence (line 159) including replacing ‘higher’ with ‘greater’ so that it now reads “*The variance in expression levels is, though, greater among those individuals carrying regulatory variant sets linked to higher expression of this gene*”. So those individuals with higher predicted expression levels generally show greater variance in their observed expression levels than those with lower predicted expression.

- line 149: the subheading 'Haplotype-dependent variance in expression' is confusing because you tested 258 genes of which only 41% with haplotype-dependent variance in expression; only a proportion of the claimed epistatic interactions involved genes within the 41%

Thanks, we have now changed this subheading to “*Interactions between cis-regulatory variant sets and the expression of distal gene networks*”

- lines 214-215: not sure if it is appropriate to test against randomness here as the 59 genes were highly selected. It is perhaps better to question here why the other ~50% did not interact, due to low power or no replication or something else?

Genotype or variant set dependent variance in expression can potentially be attributable to interaction with a variety of factors, both genetic and environmental, and it is perhaps therefore not surprising that some of those genes with evidence of veQTL effects did not have an interaction detected in our analysis. But what we do show is they are substantially enriched with those showing evidence of an interaction with the expression of another gene, which is both reassuring and supports the idea that prioritising genes displaying variant-set dependent variance in expression may be a viable approach to reduce the testing burdens further for this type of analysis.

- lines 264-267: this is an important point saying the advantage of the framework, so better to be further discussed and highlighted in e.g. Discussion

Further commentary on the principal advantages and disadvantages of the framework has been provided, for example lines 386-393.

- lines 300-301: why would be the case given your samples are likely healthy in both cohorts?

Apologies this is something we could have made clearer. We believe a large number of the interaction networks are attributable to the process of immortalisation of the cell lines using the Epstein-Barr virus. EBV causes approximately 200,000 cases of cancer annually and its immortalisation of cells has been exploited to generate cell lines. So although the individuals may have been healthy, the cell lines represent, in part, a pseudo-tumour state. This transformation of B lymphocytes by EBV has been used in various studies as an experimental model for EBV-associated cancers. So these networks provide further insights into EBV induced immortalisation. Importantly these results also suggest that for example the veQTLs identified in previous studies which largely overlap the ones we identified, likely reflect interactions resulting from this cellular transformation rather than necessarily interactions existing in the original individuals. We have now added a section to the introduction making this link between EBV transformation and cancer clearer at lines 88 to 94 and further text in the results and discussion at lines 266 to 298 and 364-393.

- line 318: give reference for "Zhernakova et al."

We have now added this at line 377. Thanks.

- line 581-582: P values were zero in Figures 5c and 5d, typo?

Thanks, this is corrected in the revised figures.

Reviewer #3 (Remarks to the Author):

Manuscript number: NCOMMS-18-7184855

Title: A novel approach for defining gene-gene interactions identifies networks of genes linked to cell immortalisation.

In this manuscript Wragg et al, perform a gene-gene interaction (GxG) analysis in LCLs using the genetic contribution of gene expression as an aggregate of the cis-variants in haplotypes instead of the gene expression measurements. The authors also performed a veQTL analysis, since genes exhibiting variance effect are influence by either GxG interactions or genotype-by-environments (GxE) interactions. By this, the authors further tested interactions only for the genes with significant veQTLs effects. This approach allows reducing the multiple testing burden of standard GxG analysis with a large searching space while it aims to capture the total genetic contribution of an individual genotype to gene expression. As a result, the authors identify GxG interactions in expression from a microarray study (LBC1936), replicating many of their finding in the GEUVADIS dataset.

The manuscript is very well written. The authors manage to explain complex concepts in a way that is easy to understand and their work is relatively easy to follow. I am positive about the work and consider that it is an interesting addition to the recent studies using similar approaches to identify non-additive genetic effects. Their solution is however not completely novel, but I believe it is important to show how the combination of genetic and variance effects can be used to identify trans associations. Please note that the use of predicted expression from the genetic component to identify GxG interactions has been presented by the authors of PrediXcan in a recent preprint (Wheeler et al, BioRxiv 471748).

There are number of issues, one of which I believe may influence the results. In addition, some of the conclusions presented in the discussion are poorly supported by the results and need some clarifications:

1) Regarding the LBC1936 dataset, the authors need to explain how they treated the difference probes during their multiple testing corrections. Multiple probes report expression levels for the same gene, giving highly correlated phenotypes. Those correlations influence the distribution of the pvalues and therefore the multiple testing corrections. Hopefully, an adequate multiple testing correction will not modify much the list of genes included giving the authors are only testing 258 genes for gene-gene interactions, but an appropriate correction is needed here and it will influence the list of genes provided as a result.

As the reviewer highlights, if the multiple probes for a gene are redundant then this may be impacting the FDR calculation if it is skewing the P value distribution. However other approaches that have been used to mitigate probe redundancy, for example averaging across probes, taking

the first principal component or adjusting the P value of the top probe to account for multiple testing potentially have issues. This is because during the array design multiple probes were selected for some genes because of the difference in expression of its isoforms and we observed that many probes for the same gene are not positively correlated.

For example, Supplementary Figure SF10 illustrates two probes for the same gene showing little correlation and two further probes in fact showing a negative correlation. Thus many probes are reporting the expression of different isoforms of the same gene, and therefore approaches for removing this redundancy, come with the risk of potentially increasing noise or missing isoform effects. This issue has previously been reported by Stalteri and Harrison (2007; doi:10.1186/1471-2105-8-13) with regards to the Affymetrix GeneChip, who found that individual probes map to more than one transcript dependent upon the biological condition.

We agree with the reviewer though that multiple probes could be skewing the P value distribution and consequently the FDR values. To investigate if this is the case we re-calculated FDR values separately for gene pairs that involved genes with multiple probes, and for gene pairs that only involved genes with a single probe. If the FDR values of the single probe genes changed after excluding the multi-probe genes, then this would suggest that these multi-probe genes may indeed be biasing the results. However as shown in Supplementary Figure SF10 the FDR values were indistinguishable suggesting retaining the multiple probes were not skewing the P value distributions or FDR results. Maintaining the multiple probes therefore allows for isoform effects to be detected without skewing the FDR values. We have added a discussion of this along these lines at lines 490 to 502. Many thanks for highlighting this.

2) Most of the discussion, and the title of the manuscript, make reference to the network associated to LDHC. From a network of GxG interactions, the authors link their work to the LCLs transformation process and to cancer. This is completely speculative and not supported by the results. Are the authors implying causality here? Because of this, I find the title misleading. They have no evidence for the implication of the GxG interaction network to cell immortalisation, only of the relationship between the genes and the genetic component of LDHC. The fact that many of those genes have been implicated in immortalization is a supporting point for the hypothesis that the network itself is involved, but at this point, and as presented in the manuscript, this is only a hypothesis that the authors are not actually testing any further. On the other hand, the two larger networks have in common their relationship with metabolism, and the GO enrichment analysis showing “ATP synthesis coupled electron transport” and “Oxidative phosphorylation” seem to support this more strongly than the cancer associations the authors are presenting. Unless the authors can provide stronger evidence for the link of the network with cell immortalisation, these strong statements need a revision.

Following the suggestions of reviewer 4 we redefined the interactions after using PEER. This led to more interactions being defined than previously which were observed to be even more highly enriched with cancer associated terms than before. We go into this in more detail at lines 267 to 289 but for example a top term was genes downregulated in nasopharyngeal carcinomas, a form of cancer associated with the EBV transformation of epithelial cells (e.g. see Young and Dawson, Chin J Cancer 2014). Some of the other most enriched terms included genes linked to the BRCA1 tumour suppressor and MYCN oncogene and we have now added figure 6 to the main manuscript

listing the most enriched terms in each network which illustrates the strong enrichment of cancer associated terms beyond just the LDHC network.

As the enriched terms pointed towards a strong association with EBV induced immortalization we have now also used ChIP-seq data to investigate the binding of EBV transcription factors at the promoters of interacting genes. EBV nuclear antigens are vital for the transformation of cells by the virus and we observe that interacting genes are substantially enriched with the binding of EBNA1P that is known to remove transcriptional repressors to help induce immortalization. We show that the genes are also preferentially bound by NF- κ B subunits that are also induced in immortalized cells to maintain proliferation. Consequently, following the reviewer's suggestion, we have now built up the evidence linking these interacting genes to cell immortalization.

3) The correlation analysis is slightly confusing. First, the authors need to explain why is it relevant to perform a correlation analysis between observed expressions, while the main advantage of the approach presented here for gene-gene interactions is using predicted expression. What is correlation when using the same phenotypes used in the interaction analyses?

Apologies we have tried to improve the clarity of this section (lines 217-219). Gene coexpression has often been used to try and identify pairs of genes potentially falling within the same network. We were just trying to illustrate that many of these interactions would not be detected using this approach, i.e. if one just looked at the coexpression between the two genes.

Second, the main conclusion presented as a consequence of the analysis is "that restricting the analysis to just co-expressed genes as in previous studies would have potentially missed a large number of the putative interactions identified in the current study.". This needs a citation (Is it maybe Zhernakova et al, 2017?) and an explanation, since it is the first time the authors are suggesting it is common practice to subset genes for gene-gene interactions analyses using correlations. In addition, this argument is contradictory with the main premise of the manuscript. If testing only correlated genes misses a large number of putative interactions, then testing only genes with veQTLs, as the authors have done here, have also the potential to miss a large number of putative interactions. The authors seem to criticize an approach similar to theirs without admitting their own limitations in the matter.

We have now included the reference to Zhernakova et al. (2017). In our analysis we hadn't actually just restricted to genes showing veQTL effects. Rather, for the reason the reviewer raises, we had tested all genes but then explored how restricting to just those genes that showed veQTL effects may have enriched for interacting pairs. We showed this would indeed substantially enrich for the interactions but as the reviewer suggests a subset would have been missed (see lines 227 to 239). We have now added further text highlighting that although many interactions were detected there are indeed limitations of this approach, for example genes with no strong cis-regulatory effect are largely inaccessible in this analysis (lines 381 to 393).

4) Please cite the R packages or any other software used. For example, what package was used for the implementation of the GLM models?

All software and packages are now cited in the text e.g. see lines 454, 455, and 488.

Minor comments:

- Page 10, line 258 = write range of values for variance explained.

Standard deviations have now been added to the variance values presented in lines 302 to 311.

- Page 7, line 180: I believe the authors mean “testis” instead of “testes”.

Many thanks. We have now changed this (line 349).

- In the manuscript, the RNAseq dataset is named “1kG cohort”, “1kG dataset”, or even the more common consortium name that produced the data “GEUVADIS”. The authors need to choose one name and be consistent through the manuscript to avoid confusing the reader.

Thanks. ‘1kG’ has now been replaced with ‘GEUVADIS’, and ‘cohort’ for ‘dataset’ throughout.

Reviewer #4 (Remarks to the Author):

The manuscript “A novel approach for defining gene-gene interactions identifies networks of genes linked to cell immortalization” by Wragg et al., describes a new framework for testing for non-additive interactions between a collective set of genetic determinants of cis regulatory effects on expression of a given gene and the expression levels of distal genes, such as transcription factors. This approach tests the interaction of a set of regulatory variants as a single component (what they term cis-regulatory haplotypes) identified using predictive models, such as PrediXcan, with the expression levels of other genes in the genome, as opposed testing each regulatory variant (eQTL) individually. This enables to decrease the multiple hypothesis burden and to potentially increase power of detection. In addition to providing new biological insight into the molecular mechanisms of gene regulation, the authors show that such interactions can explain a substantial amount of the unexplained variation of gene expression differences between individuals (4-10%) for a subset of eQTLs. The authors test their framework on 876 lymphoblastoid cell lines (LCLs) from a Scottish birth cohort, and replicate their results in an independent set of LCL samples from European individual from the 1000 Genomes project. Using a general linear regression model, the authors identify 38 genes out of an initial set of about 700 genes with significant interactions between the cis genetic determinants of their expression levels with the expression of distal genes. They show that these genes are involved in ATP synthesis that enables immortalized cells to rapidly proliferate.

This is the first time to that interactions between a set of variants controlling gene expression and the expression of another gene/trans factor are systematically tested. While the approach and methodologies used are solid and the results are of broad interest, I have several comments/concerns about the normalization and covariate correction applied to the expression data, and the evaluation of this approach compared to the typical approach of testing interactions between individual variant eQTLs versus a set of regulatory effect with the expression of other genes. I also provide a few suggestions for clearer presentation of the results.

Major comments:

1. From the analyses in the paper, it is not clear to what extent using the regulatory haplotypes that collectively explain a larger proportion of expression variation of a given gene, increases power of detecting non-additive gene-gene interactions relative to individual genetic regulatory effects on expression. Can the authors compare their non-additive interaction results using regulatory haplotypes to that using individual variant effects (eVariants), e.g., using the most significant eQTL per gene_A or testing individually each of the independent eQTLs per gene? Testing for interactions with individual eQTLs might be advantageous when there are several haplotypes with different or opposite effects on the expression levels of a given gene and only one of the haplotypes interacts with a distal gene.

Many thanks for the comment. We have now carried out the analysis the reviewer suggested. As shown in supplementary Figure SF2 using the set of cis-regulatory variants identifies ~ 30% more significant concordant interactions compared to just using the most significant eQTL. As mentioned in a previous response above, despite the larger number detected using the variant set approach, a subset of the interactions were though only detected using the single best eQTL. This is likely, at least in part, due to the reasons the reviewer highlights and we have now added a list of significant concordant interactions identified using this approach as Supplementary Table ST17 and a discussion of this at lines 386 to 393.

2. The authors use the expression 'regulatory haplotypes' to represent a set of variants that collectively contribute to the gene expression variation. I feel that this terminology should be changed as it is not precise. Haplotypes refer to a set of alleles that lie on the same chromosome, usually identified through statistical phasing of genotype data or long read sequencing. The regulatory haplotypes referred to in the paper are a set of variants whose genotypes are associated with gene expression levels, however, they may be on separate haplotypes. I would recommend using a different term, such as 'regulatory variant sets'.

This has now been changed to "regulatory variant set" throughout the manuscript.

3. On page 14, in the methods section, quantile normalization is applied to the array data to align the distributions of all probes/genes between samples. It is also important however that the probe or gene expression levels are normally distributed across samples for the performance of the linear regression model employed in PrediXcan and possibly for the general linear models used to test for non-additive interactions. The authors mention this in the Discussion on page 11, line 275, and state that they attempt to address this through their replication scheme. Have the authors checked the distribution of expression of each gene across samples? A common normalization approach to ensure each gene's expression levels are normally distributed across samples is standard normalization (e.g., as done in GTEx, PMID: 29022597) or log transformation. I would recommend the authors check how applying standard normalization to the expression data from the birth cohort affects their results.

This is also expected to reduce the effect of outliers.

We agree with the reviewer and genes with high kurtosis were excluded to minimise this effect (see line 486). The data for most genes were approximately normal but we had not applied a more harsh normalisation due to previous studies illustrating that these approaches can lead to false positive interaction effects and which recommended limiting data transformations (Castaldi et al, PloS One 2017). So our approach is an attempt at a balanced compromise between minimising the transformation of the data to reduce the introduction of false positives while excluding genes with a heavy tail so that genes with outliers are excluded. We believe one of the advantages of replicating across technologies is that the gene needs to survive these filters when measured using

both approaches and although the presence of some false positives can't be excluded the strong enrichment of relevant biological terms among the interactions is reassuring.

Also, on page 14, can the authors briefly mention how the gene expression data for the 344 European individuals from the 1kG dataset were normalized.

See response directly below.

4. In the general linear model of the observed gene expression of a given gene (in Methods, page 15), the authors adjusted for sex, age, and population stratification using the first five genotype principal components (PCs). However, they do not adjust for potential batch effects or hidden experimental or other confounding factors on gene expression variation, which have been shown in many studies to exist. The authors note at the end of the Results section on the bottom of Page 10, that some of the discrepancies between the two studies, the birth cohort and 1KG, could be due to technical variation in gene expression measurements in the array study vs. the RNA-seq study. Did the authors choose not adjust for hidden covariates in expression data as they were concerned that this would remove true trans effects on expression variation? I would suggest testing how the number of non-additive interaction changes with adding in hidden expression covariates, for example using the Probabilistic Estimation of Expression Residuals (PEER) method (PMID: 22343431), as applied for example for the trans eQTL analysis in GTEx (PMID: 29022597).

As the reviewer suggests we did have concerns about using approaches such as PEER which remove hidden factors explaining expression variability irrespective of their underlying cause. As stated in the original PEER paper these factors can be related to pathways or transcription factor activations which we would not want to remove in our analysis. We have therefore tried to come up with a balanced approach that mitigates the potential confounding by batch effects while not removing these true biological effects.

For the LBC1936 dataset we have now explicitly accounted for technical variation (sample plate and sentrix assay ID) as random effects, and regressed these out in addition to sex and age in all analyses. For the GEUVADIS data, processing laboratory and assay ID were accounted for in the same way. When generating our prediction models we have now also used PEER as suggested by the reviewer to remove non-cis genetic factors impacting expression variability. This has had the result of increasing the number of genes with good prediction models, as may be expected due to the non-cis genetic variability being removed. However, when testing for interactions between these predicted expression values of a gene and the observed expression of a second gene the response of the model was the expression level of the gene having accounted for plate effects but without applying PEER. This therefore allows us to identify interactions explaining expression variability that would otherwise be removed by PEER. We believe this approach gives us the best compromise between accounting for batch effects and improving the predictions models while not removing expression variability linked to the biology.

The effect of these changes has been an increase in the number of genes with good prediction models and an associated increase in the number of interactions identified. Enrichment P values have also generally got substantially smaller.

The improvement in predictions when applying the LBC models to the GEUVADIS expression values before (lmer.1kG) and after (peer.1kG) using PEER.

Regarding the GEUVADIS dataset, expression values normalised for library depth and transcript lengths (RPKM) were downloaded (https://www.ebi.ac.uk/arrayexpress/files/E-GEUV-1/analysis_results/). We also sourced the metadata for the samples used in the study from EMBL-EBI's ArrayExpress database (<https://www.ebi.ac.uk/arrayexpress/experiments/E-GEUV-1/samples/>). We processed this data as described above.

The kurtosis filter was applied to all four datasets (LBC1936 observed and predicted expression, GEUVADIS observed and predicted expression). Only genes that did not exhibit excess kurtosis in any of the four datasets were retained.

5. The authors argue in the Introduction and in their simulated example in Figure 3 that detecting variance eQTLs suggests that there may exist non-additive interactions between the eQTL and the expression levels of a transcription factor. However, the authors do not investigate this hypothesis further in the interactions they find between the regulatory haplotypes and the expression of distal genes. For example, are the distal genes found to interact with the genetically determined expression levels of a given gene enriched for known transcription factors or other regulators?

We did observe that the distal genes are strongly enriched with the positive regulation of gene expression GO term (adjusted $P = 8.0 \times 10^{-8}$) suggesting they are preferentially associated with expression regulation, however we should have clarified this statement as we don't believe that TF binding directly to the eQTLs is necessarily the only potential explanation. Upstream factors in the same pathway may also show an interaction even if they don't directly bind to the eQTL. Also genes and their targets that regulate chromatin are potential candidates, and the association with EZH2 is potentially reflective of this. We have therefore made changes at lines 135 and 371 to clarify this.

Furthermore, on page 7 in the Results section, the authors note that a large proportion of the 244 high confidence interactions found are centered on a common set of genes. Based on the hypothesis that transcription factors may underlie much of the non-additive interactions, I would have expected to see an opposite pattern, i.e. interactions between a single trans regulator (distal gene_B) and multiple target genes (gene_A). Do any of the interactions show such a pattern? Why are we not seeing this pattern as the more common pattern?

Also, can the authors note how many target genes, i.e. 'gene_A' comprise these 244 interactions and show the distribution of number of interactions per gene_A.

See response directly above regarding the first point and transcription factors. We have now also included Supplementary Figure SF3 illustrating the number of interactions per gene^A.

6. To make it easier to follow all the results in the paper I would recommend summarizing the main findings in a table, e.g., number of genes with a significant PrediXcan models, number of significant veQTLs, number of significant GxG interactions, intersections, etc., with a column for significance in the Scottish birth cohort, 1KG and in both.

As recommended, we have now included a summary of counts for the above steps. These are provided in Supplementary Table ST7 and cited in the text at line 184.

7. Such interactions between distal genes and the genetic determinant of gene expression of another gene were found for only 20% or less of genes tested. Can the authors comment on possible reasons this may be.

This figure is now closer to a third of tested genes in the updated analysis. We are not sure you would necessarily expect to see all genes involved in such a statistical interaction for various reasons. One example is even if the prediction models were perfect, the transcript levels of the geneBs will not necessarily represent their protein levels. So if for example a TF protein does bind to a gene's regulatory site its transcript levels as measured by RNA-seq may still not show an interaction with the regulatory site's genotypes. We have now added this limitation to the discussion section at lines 382 to 393.

Minor comments:

1. In Figure 1b, the model of the observed expression levels of a given gene is shown only as a function of an interaction term between the genetically determined component (in cis) of gene expression of this gene and the observed expression of another distal gene, but it should also include these effects individually and other potential effects (e.g., environmental effects not related to the distal gene's expression levels).

We have now added an error term to this figure, that represents the model we were fitting, and have described in the legend how this term will include, for example, uncaptured environmental effects.

2. In the introduction, on page 3, can the authors add a reference to support the sentence on lines 49-50: "Each cis-eQTL has, though, been found to generally only explain a small proportion of the variation in a gene's expression between individuals.", e.g., Gamazon, E. R. et al. A gene-based

association method for mapping traits using reference transcriptome data. Nat. Genet. 47, 1091–1098 (2015).

Reference now added, thanks.

3. On page 5, lines 116-117, can the authors add a reference to the sentence: “Previous studies have highlighted how both genetic and environmental interactions can lead to genotype dependent variance in a gene’s expression.”

References added, many thanks

4. On page 7, in the sentence on lines 178-179, for further clarification I might add the word ‘trans’ to the ‘untested factor’ (i.e. untested trans factor driving the pathway).

Many thanks. We have added this.

5. On page 8, line 194, is the R^2 a Pearson or Spearman correlation of determination?

We have now clarified this at line 213. Thanks.

6. In the methods section on page 15, it says that only genes reported by PredictDB as having a significant prediction model were trained with the PrediXcan model in the Lothian birth cohorts. Why did the authors not test all genes, since the model significance should be dependent on the study in which it was trained.

PredictDB is used to train the models whereas PrediXcan is used to makes predictions from them. We did train models in the LBC1936 cohort for all genes that were not filtered e.g. due to excessive kurtosis, but the models for some genes were observed to have no predictive value. As the predictions from these models were not informative we did not take them forward.

The same models trained using predictDB using the LBC1936 data were then applied to the independent GEUVADIS cohort to assess the reproducibility of these models. Although the reviewer is correct in that we would expect models separately trained in the GEUVADIS cohort to have higher prediction accuracies in the same dataset this wouldn’t strictly be an independent replication. Applying the LBC1936 models to the GEUVADIS dataset, where expression values were assayed using a different platform, is expected to be the more conservative approach.

7. In the methods section on the bottom of page 17, can the authors briefly state what statistical test is used in ConsensusPathDB to assess gene set enrichment in the given subnetworks.

We have updated this enrichment analysis in the new manuscript using FUMA (<https://fuma.ctglab.nl/>) which uses the hypergeometric test. We have updated the description of this analysis in the methods at lines 613 to 624.

8. I would recommend increasing the font of the genes in Figure 6, at least those in the hubs.

We have now replaced this figure.

9. In figure 8, are the units of the y-axis in panels A, B, and C percentage or fraction? Based on the results I assume it is percentage?

Thanks. We have now clarified in the figure 8 legend that these are proportions of variance explained.

REVIEWERS' COMMENTS:

Reviewer #1 (Remarks to the Author):

I am happy that all of my questions have been addressed apart from comment #4. There was insufficient clarity as to why the authors persisted in using microarray datasets and did not incorporate RNA-Seq data.

Reviewer #2 (Remarks to the Author):

I think the authors have done an excellent job in this revision. All my comments have been addressed satisfactorily. I have no further comments.

Reviewer #3 (Remarks to the Author):

All points and comments have been address by the authors. This reviewer has no further questions or comments.

Reviewer #4 (Remarks to the Author):

The authors of the manuscript "A novel approach for defining gene-gene interactions identifies networks of genes linked to cell immortalization" have satisfactorily addressed my comments and concerns and I think the manuscript is close to publication. I only have two final comments:

1. In Supplementary Table 17 – can the authors add the variant IDs of the most significant/lead eQTL variant tested per gene in the interaction analysis.
2. In response to reviewer #3's comment #1, the authors test how having different numbers of probes on the array per gene may be affecting the multiple hypothesis correction, by calculating FDR values separately for gene pairs that contained multiple probes, and for gene pairs with a single probe (results shown in Supplementary Figure SF11). First, the axis labels in Supplementary Figure SF11 are a bit confusing, as based on the Supplementary figure title (Supplementary Figure SF11. FDR values of single probe genes before and after excluding pairs involving multiple probes), I would expect one axis to be labeled "FDR across pairs after excluding pairs involving multiple probes", but the axes are "FDR across pairs involving multiple probes" and "FDR across all probes". More importantly, the results between the two axes look identical, which is a bit surprising/concerning. I would expect some deviation from the diagonal. Can the authors double check that indeed the results are correct. Finally, it is not fully clear from the text that was added in the methods on lines 490-502 addressing this point, what model the p-values refer to. Is it from equation 2?

RESPONSE TO REVIEWER

Reviewer #4

1. In Supplementary Table 17 – can the authors add the variant IDs of the most significant/lead eQTL variant tested per gene in the interaction analysis.

Additional columns have now been included in (what is now) Supplementary Data 15 to report the rsid associated with the best eQTL for geneA in both the LBC1936 and GEUVADIS datasets. The most significant eQTL variant tested per gene in the interaction analyses are also presented in Supplementary Data 13 and 14 for LBC1936 and GEUVADIS datasets, respectively.

2. In response to reviewer #3's comment #1, the authors test how having different numbers of probes on the array per gene may be affecting the multiple hypothesis correction, by calculating FDR values separately for gene pairs that contained multiple probes, and for gene pairs with a single probe (results shown in Supplementary Figure SF11). First, the axis labels in Supplementary Figure SF11 are a bit confusing, as based on the Supplementary figure title (Supplementary Figure SF11. FDR values of single probe genes before and after excluding pairs involving multiple probes), I would expect one axis to be labeled "FDR across pairs after excluding pairs involving multiple probes", but the axes are "FDR across pairs involving multiple probes" and "FDR across all probes". More importantly, the results between the two axes look identical, which is a bit surprising/concerning. I would expect some deviation from the diagonal. Can the authors double check that indeed the results are correct. Finally, it is not fully clear from the text that was added in the methods on lines 490-502 addressing this point, what model the p-values refer to. Is it from equation 2?

The axis labels on Supplementary Figure 11 have now been corrected, and the plotted values checked. The FDR results before and after excluding genes with multiple probes are very similar but not in fact identical. To illustrate this the diagonal (intercept = 0, slope = 1) has been added to the figure, and a further panel (11b) included to better illustrate the deviance from the diagonal. This analysis is based on Eqn 3, which is now clarified in the text at line 510.